# PrivateChat: A Secure Encrypted Communication Framework with Black-box LLMs

## Abstract

With the growing applications of large language models (LLMs), privacy leakage has emerged as a significant concern. However, widely used LLMs are often deployed on cloud platforms and accessible only through relatively expensive API calls, complicating the realization of secure communication between users and cloud LLMs. In this paper, we introduce **PrivateChat**, a novel private communication framework that enables users to safely interact with cloud LLMs using user-customized encryption methods (e.g., *AES*). Our core idea is to learn a private system prompt, which instructs the cloud LLM to process and respond in encrypted text while concealing encryption details from potential attackers. Additionally, to optimize such prompts with few API calls, we propose a Sample-Efficient Simultaneous Perturbation Stochastic Approximation (SE-SPSA) black-box optimization algorithm, which incorporates a baseline-based variance reduction strategy with SPSA for effective and economical training. Extensive experiments on several benchmark datasets with various encryption methods show the effectiveness of our approach in achieving secure and reliable communication with cloud LLMs.

## 1 Introduction

In recent years, large language models (LLMs) have been extensively applied in various tasks, such as text generation, language translation, and question answering. However, these LLM applications (e.g., GPT-4 (OpenAI, 2023b) and Claude (Anthropic, 2023)) are often deployed on cloud platforms (i.e., cloud LLMs), posing risks of private information exposure to hackers and service providers in the data transmission process. The privacy risk of LLMs manifests in two main ways: (1) Entity privacy leakage: Users might unintentionally expose their sensitive information (e.g., names, addresses, and age) in their input queries (Lukas et al., 2023); (2) Inference privacy leakage: Potential attackers could deduce personal data (e.g., health, income, and gender) through the user chat records with the LLMs, even if the input text does not explicitly contain private details (Staab et al., 2023). These privacy risks limit the wider applications of LLMs, and many countries have established laws and regulations to restrict and even prohibit their use (Neel & Chang, 2023).

In light of the aforementioned privacy risks associated with using cloud LLMs, secure communication methods are essential. Encryption techniques, such as those employed by communication platforms for ensuring privacy and security, serve as precedents (Lai et al., 2017). This inspires us to explore the feasibility of an encrypted communication framework tailored for interacting with cloud LLMs. This is a novel and highly encouraging research direction, which yet poses a series of new research problems. In detail, to prevent the aforementioned entity and inference privacy leaks to attackers and service providers, both the user query and the LLM's response should be encrypted during the data transmission process. However, how to enable LLMs to accurately understand and respond to encrypted texts is a non-trivial challenge. In particular, unlike the white-box assumption where the model structure and parameters are accessible, as used in previous privacy-preserving methods (Qu et al., 2021; Zhou et al., 2023), the widely-used emerging LLMs (e.g., GPT-4) are typically black-box, with closed and inaccessible model architectures and parameters. This black-box nature hinders us from directly using the prevalent back-propagation algorithm to fine-tune these black-box LLMs for processing the encrypted texts. Last but not least, even if we could adopt a black-box optimizer, such as SPSA (Spall, 1992a), to fine-tune LLMs through prompt tuning, it would consume numerous sample data for trial-and-error learning (Spall, 2000; 1997a). However,

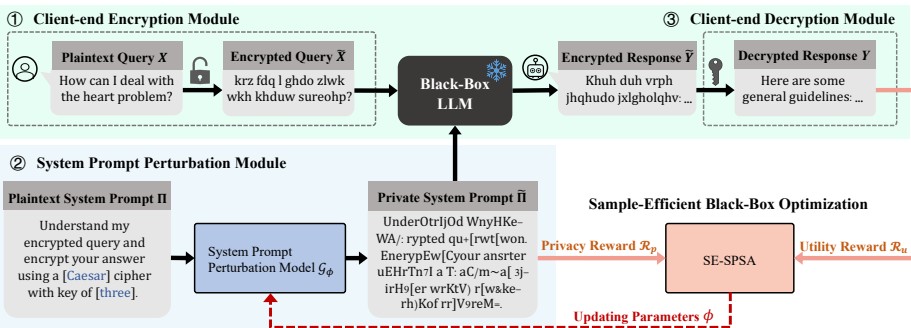

Figure 1: The pipeline of our PrivateChat framework. It enables encrypted communication between users and black-box LLMs under the guidance of a private system prompt. The framework is optimized using our SE-SPSA black-box optimizer, ensuring economical and effective learning.

in our task, training samples are derived from expensive API calls to cloud-based LLMs, making existing black-box optimizers unsuitable.

To address these challenges, we introduce **PrivateChat**, a novel private communication framework ensuring encrypted interactions between clients and cloud black-box LLMs. Our core idea is to train an effective generative model to produce high-quality *private system prompts*, safely written with encryption details, for instructing the cloud LLM to process encrypted queries while safeguarding its encryption details from potential attackers. Specifically, as shown in Fig. 1, our PrivateChat comprises three modules: the client-end encryption module, the system prompt perturbation module, and the client-end decryption module. Our client-end encryption module encrypts the user's plaintext queries using the user-customized encryption method (e.g., *AES*) and key. Subsequently, our system prompt perturbation module securely embeds these encryption details (i.e., the encryption method and the key) into a system prompt for safely guiding the cloud LLM to process the encrypted query and generate the encrypted response. Next, we submit the encrypted query alongside the private system prompt to the cloud LLM, which returns an encrypted response. Finally, the client-end decryption module decrypts this response into a user-comprehensible plaintext. Note that the generated private prompt can be conveniently reused for subsequent multi-round encrypted dialogues without regeneration. Via such a carefully-designed framework, our PrivateChat enables encrypted communication between users and cloud LLMs, effectively preserving the user privacy.

Nevertheless, it is a non-trivial task to effectively optimize our system prompt perturbation module to produce a desired prompt. First of all, the black-box nature of the cloud LLMs makes the prompt perturbation module non-differentiable, rendering prevalent back-propagation optimization nonfunctional. Moreover, although current black-box optimization methods, such as SPSA (Spall, 1992a), can estimate gradients through trial-and-error learning, such a learning paradigm typically consumes numerous training data samples. In our task, these training samples come from expensive API calls of cloud LLMs, resulting in high training costs. In this paper, these difficulties motivate us to develop a novel black-box optimizer, called Sample-Efficient Simultaneous Perturbation Stochastic Approximation (SE-SPSA), for effective and economical training. Specifically, beyond just using sample data for SPSA-based gradient estimation, we also utilize them to compute an effective baseline for reducing the variance of the gradient estimation. This strategy not only stabilizes and accelerates convergence but also significantly improves the performance by providing more accurate and reliable gradient estimates. Besides, we design two effective reward functions (namely, the utility reward and the privacy reward) as our optimization objectives to ensure both the accuracy of the LLM responses and robust privacy for the private system prompt.

To summarize, our main contributions are as follows: **1)** To protect chat content from hackers and service providers, we introduce a novel private communication framework, PrivateChat, enabling safe and encrypted interactions between users and cloud black-box LLMs. To the best of our knowledge, this is the first end-to-end encrypted communication framework between users and cloud black-box LLMs for user privacy protection. **2)** We propose a system prompt perturbation module, which generates effective private system prompts for instructing the cloud LLMs to understand and respond to queries with user-customized encryption methods. To tackle the challenges posed by the black-box nature and costly API calls of cloud LLMs during the optimization of our private prompt, we develop a new sample-efficient black-box optimizer, SE-SPSA, which incorporates a baseline-

based variance reduction strategy with SPSA for effective and economical training. **3)** Extensive experimental results on different benchmark datasets with various encryption methods including *Caesar*, *DES*, *AES*, and *ChaCha20* demonstrate the outstanding utility and privacy-preserving abilities of our framework.

## 2 RELATED WORK

**Large Language Models.** In recent years, numerous large language models (LLMs) like ChatGPT (OpenAI, 2023a;b), LLaMA (AI, 2023), and Claude (Anthropic, 2023), have been developed, showing great value in various fields, including code generation (Jain et al., 2023; Gui et al., 2024; Mu et al., 2024), healthcare (Thirunavukarasu et al., 2023; Bazi et al., 2023; Li et al., 2024; Liu et al., 2023a), education (Lee et al., 2024; Bewersdorff et al., 2024), and finance (Ionașcu, 2023; Muhtar et al., 2024). However, the cloud deployment of commercial LLMs (e.g., GPT-4) raises significant privacy concerns (Yao et al., 2024; Das et al., 2024), as user data transmitted to these services can be vulnerable to interception by hackers or misuse by service providers (Wang et al., 2023). Various attack methods further highlight the LLMs' vulnerabilities, such as bypassing LLMs' security checks to obtain sensitive information (Yuan et al., 2024), and inferring personal privacy through inference attacks (Qu et al., 2021; Dong et al., 2023). While some research efforts (Zhou et al., 2023; Liu et al., 2023b) explore privacy protection in LLM usage, they often require fine-tuning, unsuitable for black-box LLMs with closed architectures. Here, we propose PrivateChat, the first secure encrypted communication framework designed for black-box LLMs, ensuring user privacy.

**Privacy-preserving Methods.** Some techniques such as distributed computing (Qin et al., 2014), homomorphic encryption (Ibtihal et al., 2020) and federated learning (Liu et al., 2020) safeguard client data confidentiality, but they require close collaboration between the LLM and the client (e.g., exchanging model parameters and gradients). This reliance limits their applicability to cloud-based LLMs, which are typically accessible only through commercial APIs. Text sanitization is also a common privacy-preserving method, employing approaches like local differential privacy (Yue et al., 2021; Chen et al., 2023a), which adds random noise during data processing, or anonymization (Chen et al., 2023b; Vats et al., 2023; Kan et al., 2023), which masks or replaces private entities. However, these approaches inevitably incur a certain degree of utility loss (Zhang et al., 2024). Moreover, they only disrupt parts of the user input and fail to protect privacy within LLM responses, allowing attackers to infer private information from both the input context and LLM replies. Here, we are the first to propose a novel framework that enables users to interact with LLMs via ciphertext, ensuring end-to-end privacy protection (e.g., covering both user input and LLM output) without sacrificing information. Furthermore, we design a sample-efficient black-box optimizer to enhance the utility and privacy-preserving capabilities of our framework in a black-box setting.

**Black-box optimization.** Traditional black-box optimizers (Lillicrap et al., 2015; Tsai et al., 2020; Spall, 1992a) often use techniques like reinforcement learning (Lillicrap et al., 2015), derivative-free optimization (Ghanbari & Scheinberg, 2017), and one-sided gradient estimators (Tsai et al., 2020) for parameter updates. However, these methods struggle to converge in high-dimensional parameter spaces. Although the simultaneous perturbation stochastic approximation (SPSA) methods (Spall, 1992a; Oh et al., 2023) effectively estimates high-dimensional gradients, it usually leads to unstable optimization (Zhao et al., 2011), which, in our task, necessitates numerous expensive API calls to cloud LLMs, resulting in high training times and costs. Moreover, this instability complicates finding optimal solutions, limiting performance. Differently, we propose SE-SPSA, a novel sample-efficient black-box optimizer that combines SPSA with a baseline-based variance reduction strategy, stabilizing gradient estimates and improving optimization reliability and performance with reduced training times and costs.

## 3 METHOD

In this paper, we propose **PrivateChat**, a novel private communication framework for secure encrypted interactions between users and cloud LLMs. As shown in Fig. 1, our framework consists of three modules: the client-end encryption module, the system prompt perturbation module, and the client-end decryption module. Given a user's plaintext query, the client-end encryption module first encrypts it into ciphertext (Sec. 3.1 (1)). Next, the system prompt perturbation module gen-

erates a private prompt to guide the cloud LLM in processing the ciphertext query and producing an encrypted response without revealing encryption details (Sec. 3.1 (2)). The ciphertext query, along with the private system prompt, is then sent to the cloud LLM. Upon receiving the ciphertext response from the LLM, the client-end decryption module converts it back into plaintext for users to read (Sec. 3.1 (3)). Additionally, we introduce SE-SPSA, a novel sample-efficient black-box optimization framework designed to optimize our framework effectively and efficiently (Sec. 3.2).

## 3.1 PRIVATE COMMUNICATION FRAMEWORK

**(1) Client-end Encryption Module.** To prevent chat records from leaking to attackers and service providers, we encrypt user queries on the client end before sending them to the cloud LLM. To this end, we design a client-end encryption module that uses an encryption algorithm with a key to convert the user's plaintext query $X$ into ciphertext $\tilde{X}$, as shown in Fig. 1. In particular, our framework allows users to customize their preferred encryption algorithm and key, including both classical encryption algorithms such as *Caesar*, and advanced encryption methods such as *DES*, *AES* and *ChaCha20*, demonstrating its generality. Please refer to Apps. for more details on these encryption methods.

**(2) System Prompt Perturbation Module.** Upon encrypting the user's query, we send it to the cloud LLM, expecting a ciphertext LLM response using the identical encryption algorithm utilized for client-end encryption. However, it is challenging for the cloud LLM to directly understand such ciphertext query and provide an encrypted response, as it lacks knowledge of the encryption method and key required to process the ciphertext. One possible solution is to additionally submit a plaintext system prompt to explicitly inform the cloud LLM about the user-customized encryption details. However, this is unsafe, as it directly exposes sensitive encryption details, increasing the risk of privacy leakage. Therefore, our focus is to generate a safe private prompt capable of effectively guiding the LLM to process the encrypted query while concealing the encryption details.

In this paper, we design a system prompt perturbation module to generate such private system prompts. Specifically, we first design an initial plaintext system prompt $\Pi$ that explicitly instructs the cloud LLM to communicate in a user-customized encryption approach. The initial prompt $\Pi$ contains the encryption method (e.g., *Caesar*) and the user-defined encryption key, defined by the user at the client-end encryption stage (refer to Sec. 3.1 (1)). A template for this prompt is outlined below:

*"Understand my encrypted query and encrypt your answer*

*using a [**encryption method**] cipher with key of [**number or binary sequence**]"*.

Subsequently, we need to convert this plaintext system prompt $\Pi$ into a private one $\tilde{\Pi}$. The main challenge here lies in ensuring that this private prompt effectively instructs the cloud LLM (i.e., keeping utility) while simultaneously concealing the encryption details (i.e., keeping privacy), thus achieving both utility and privacy. Given the advanced contextual understanding capabilities of the LLMs, which enable them to discern the underlying semantics of heavily perturbed text (Zhao et al., 2024), we propose a learnable system prompt perturbation model $\mathcal{G}_\phi : \Pi \to \tilde{\Pi}$ to generate such private prompt $\tilde{\Pi}$ by adaptively perturbing the initial plaintext prompt $\Pi$. Here, perturbation means replacing the raw elements (e.g., characters, tokens and words) in the plaintext prompt with the codes from a pre-defined codebook.

Based on our experiments (see Tab. 2), which empirically demonstrate that both word-level and token-level perturbations significantly decrease the LLMs' performance by hindering their understanding of prompt semantics, we design a more robust character-level perturbation method. Moreover, excessive encryption, such as perturbing all characters in a plaintext prompt, also breaks semantic integrity and contextual cues, resulting in a loss of utility (see Fig. 3). To this end, our system prompt perturbation model adaptively determines which characters to perturb and how to perturb them within the plaintext prompt $\Pi = \{\pi_1, ..., \pi_N\}$ in order to generate a private system prompt $\tilde{\Pi} = \{\tilde{\pi}_1, ..., \tilde{\pi}_N\}$ that balances utility and privacy. Here, $\pi_n$ and $\tilde{\pi}_n$ represent the $n^{th}$ character in the plaintext prompt $\Pi$ and the private prompt $\tilde{\Pi}$, respectively, where $n \in \{1, ..., N\}$ and $N$ is the prompt length.

Specifically, our model comprises two types of learnable parameters: the perturbation probability distribution $P^P$ and the encoding probability distribution $P^E$. The perturbation probability distribution $P^P = \{p_n^P\}_{n=1}^N$ determines which characters in the plaintext prompt should be perturbed,

where $p_n^P$ denotes the probability of perturbing the $n^{th}$ character $\pi_n$ in the plaintext prompt. For each character $\pi_n$, encoding probability distribution $P_n^E$ determines how to perturb it, where $p_{n,r}^E$ represents the probability that the character $\pi_n$ should be perturbed as $\mathcal{C}_r$ ($\mathcal{C}_r$ denotes the $r^{th}$ code within a codebook containing a total of $R$ codes, $r \in \{1, ..., R\}$). To avoid the utility loss caused by the excessive encryption as discussed above, we just perturb the character $\pi_n$ if its perturbation probability $p_n^P$ exceeds a perturbation threshold $\varepsilon$. Via the above strategy, we produce the private prompt $\tilde{\Pi} = \{\tilde{\pi}_1, \ldots, \tilde{\pi}_N\}$ as follows:

$$\tilde{\pi}_n = \begin{cases} \mathcal{C}_{r^*}, & \text{if } p_n^P > \varepsilon, \\ \pi_n, & \text{otherwise,} \end{cases} \tag{1}$$

where $r^* = \arg\max_r p_{n,r}^E$ denotes the code index with the highest encoding probability in the codebook corresponding to $\pi_n$. Each code in the codebook is a random combination of $N_c$ ASCII characters and for simplicity, we here set $N_c = 1$. By calculating parameter gradients through feedback from the cloud LLM, we can optimize the model parameters $\phi = \{P^P, \{P_n^E\}_{n=1}^N\}$ (refer to Sec. 3.2 for detailed optimization process).

**(3) Client-end Decryption Module.** As shown in Fig. 1, after obtaining the private system prompt $\tilde{\Pi}$ generated by our prompt perturbation module, we submit it along with the ciphertext queries $\tilde{X}$ to the cloud LLM, and then the LLM can generate a ciphertext response $\tilde{Y}$. Finally, taking the generated ciphertext response $\tilde{Y}$ as input, the client-side decryption module utilizes the corresponding decryption rules, based on the user-customized encryption method (e.g., *AES*) and key, to convert the encrypted response $\tilde{Y}$ back into the plaintext response $Y$ for the user to read.

## 3.2 Sample-efficient Black-box Optimization Framework

Utilizing the private communication framework described above enables us to establish secure encrypted interaction between users and cloud LLMs. Within this framework, the generation of effective private system prompts is achieved by training our prompt perturbation model with an optimization objective, which ensures both the privacy and utility of the prompts. However, direct optimization of this objective function using the prevalent gradient back-propagation algorithm is impractical due to the inaccessible architectures of the cloud LLMs (e.g., GPT-4). While traditional black-box optimization methods can estimate gradients by extensively exploring the parameter space, they typically require numerous samples. In our scenario, this would lead to expensive API calls to LLMs, thereby making them inappropriate for our task due to their resource-intensive nature. To achieve a user-friendly system for generating optimal prompts with reduced training time and cost, we propose a sample-efficient black-box optimizer, that enables users to create private system prompts efficiently and economically. Next, we elaborate on our optimization objective and the sample-efficient black-box optimizer.

**(1) Privacy Reward and Utility Reward-based Optimization Objective.** Our training framework aims to learn an effective private system prompt that guides the cloud LLM to produce highly accurate responses (i.e., utility), while also concealing the encryption details (i.e., privacy). We thus design a utility reward function $\mathcal{R}_u$ to assess the accuracy of LLM response, and a privacy reward function $\mathcal{R}_p$ to evaluate the privacy level of the learned system prompt. These two reward functions are combined as the optimization objective to train our system prompt perturbation model $\mathcal{G}_\phi$.

**Utility reward.** The utility reward $\mathcal{R}_u$ aims to assess the accuracy of the ciphertext responses $\tilde{Y}$ from the cloud black-box LLM. The response accuracy is measured by $Rouge$-1 (Lin, 2004), denoted as $\mathcal{F}_{Rouge_1}$, which calculates the similarity between the groundtruth response $Y_{gt}$ and the decrypted response $Y$ from the cloud LLM:

$$\mathcal{R}_u(Y, Y_{gt}) = \mathcal{F}_{Rouge_1}(Y, Y_{gt}). \tag{2}$$

**Privacy reward.** The privacy reward $\mathcal{R}_p$ evaluates the privacy level of the generated private system prompt. Based on the fact that a larger difference between the private system prompt $\tilde{\Pi}$ and the original plaintext system prompt $\Pi$ tends to conceal more privacy information (Qu et al., 2021), we adopt this difference to measure the privacy degree. Specifically, we quantify this difference at both semantic and character levels. Following Sentence-BERT (Reimers & Gurevych, 2019), we calculate the semantic-level difference $\mathcal{F}_{sem}$ based on the cosine similarity $Cos$ between the BERT-based

(Devlin et al., 2018) semantic embeddings of these two prompts (i.e., $\mathcal{F}_{sem}(\Pi, \tilde{\Pi}) = \frac{1-Cos(\Pi,\tilde{\Pi})}{2}$). Also, we measure the character-level difference $\mathcal{F}_{char}$ based on the overlap rate $\mathcal{F}_{overlap}$ between the characters of the private and plaintext system prompts (i.e., $\mathcal{F}_{char}(\Pi, \tilde{\Pi}) = 1 - \mathcal{F}_{overlap}(\Pi, \tilde{\Pi})$). Since the critical parts of the system prompt we aim to protect are the encryption details (i.e., the encryption method and key), we further calculate both the semantic and character-level differences between the encryption details in the private and plaintext prompts. Thus, the total privacy reward $\mathcal{R}_p$ can be written as:

$$\mathcal{R}_p(\Pi, \tilde{\Pi}) = \mathcal{F}_{sem}(\Pi, \tilde{\Pi}) + \mathcal{F}_{char}(\Pi, \tilde{\Pi}) + \mathcal{F}_{sem}(\Pi_e, \tilde{\Pi}_e) + \mathcal{F}_{char}(\Pi_e, \tilde{\Pi}_e), \quad (3)$$

where $\tilde{\Pi}_e$ and $\Pi_e$ represent the encryption details portions of the private prompt and the plaintext prompt, respectively.

In summary, the overall objective function $\mathcal{R}(\phi)$, composed of the utility reward and the privacy reward, can be formulated as:

$$\mathcal{R}(\phi) = \mathcal{R}_u(Y, Y_{gt}) + \mathcal{R}_p(\Pi, \tilde{\Pi}). \quad (4)$$

By maximizing this objective function, we can obtain a private system prompt that ensures privacy and utility. Next, we elaborate on how to optimize this objective function using our carefully-designed sample-efficient black-box optimization algorithm.

**(2) Sample-efficient Simultaneous Perturbation Stochastic Approximation (SE-SPSA).** To optimize the objective function in Eq. 4, we need to compute the gradients for updating parameters in our system prompt perturbation model so that it can generate effective and private system prompts that maximize the utility and privacy rewards. Since the calculation of the utility reward requires feedback from the cloud black-box LLM, the gradients associated with reward need to be propagated back through the LLM. However, this process is infeasible due to the closed architecture of the black-box LLM. Thus we need to develop a black-box optimizer to estimate parameter gradients through trial-and-error learning. Nevertheless, existing black-box optimizers such as Simultaneous Perturbation Stochastic Approximation (SPSA) (Spall, 1992a), typically consume numerous samples, which are derived from expensive API calls to LLMs in our task, thereby leading to high training costs and time. To handle this challenge, we develop a novel Sample-Efficient SPSA (SE-SPSA) method for effective and economical black-box optimization. In the following, we first introduce SPSA (Spall, 1992a) and then elaborate on our new variant, SE-SPSA, which incorporates a baseline-based variance reduction strategy to stabilize and accelerate the optimization process and improve model performance.

**Simultaneous Perturbation Stochastic Approximation (SPSA).** Due to the black-box nature of the cloud LLMs, it is infeasible to leverage the back-propagation algorithm to directly compute the analytical gradients of parameters $\phi$ for optimizing our system prompt perturbation model $\mathcal{G}_\phi$ using stochastic gradient descent. Therefore, we employ SPSA (Spall, 1992a; 1997b), a black-box optimization method, to estimate the parameter gradients for model optimization. SPSA estimates gradients by randomly perturbing the model parameters $\phi$ and calculating output differences at these perturbed points. Specifically, at each optimization step, SPSA applies random positive and negative perturbations to the model parameters, measures the differences in the objective function values, and then uses the average of these differences for gradient estimation, termed as $\hat{g}_i^{spsa}$, which can be formulated as:

$$\hat{g}_i^{spsa}(\phi_i) = \frac{1}{J} \sum_{j=1}^{J} \frac{1}{\mathbf{u}_i^{(j)}} \left( \frac{\mathcal{R}(\phi_i - c_i \mathbf{u}_i^{(j)}) - \mathcal{R}(\phi_i + c_i \mathbf{u}_i^{(j)})}{2c_i} \right), \quad (5)$$

where $i \in \{0, ..., I-1\}$ denotes the optimization step ($I$ is the total number of steps); $\phi_i$ are the parameters of the system prompt perturbation module in the $i^{th}$ step; $\mathcal{R}(\cdot)$ is our objective function in Eq. 4; $c_i$ is the perturbation coefficient. Following (Oh et al., 2023), $\{\mathbf{u}_i^{(j)} = [u_{i,1}^{(j)}, \cdots, u_{i,M}^{(j)}]\}_{j=1}^{J}$ represent a set of randomly sampled perturbation vectors, where $J$ represents the number of samples and $M$ denotes the dimension of these vectors (i.e., the dimension of the flattened model parameters $\phi$). Each vector element $u_{i,m}^{(j)}$ follows a segmented uniform distribution (Spall, 2005; 1992b), specifically $u_{i,m}^{(j)} \sim 0.5 \cdot U(0.5, 1) + 0.5 \cdot U(-1, -0.5)$. With the estimated gradient $\hat{g}_i^{spsa}$, the parameter update in the $i^{th}$ step of SPSA is written as:

$$\phi_{i+1} = \phi_i - a_i \hat{g}_i^{spsa}(\phi_i), \quad (6)$$

where $a_i$ is the learning rate for the $i^{th}$ optimization step and $\phi_0$ denotes the initial model parameters.

**Baseline-based Variance Reduction.** While SPSA can help estimate parameter gradients under the black-box setting, our experiments (see Fig. 5) empirically show that SPSA suffers from limited training stability and slow convergence in model optimization, which is also observed in previous works (Oh et al., 2023; Spall, 2000). This instability stems from the stochastic nature of the *randomly* sampled perturbation vectors $\mathbf{u}_i$ used in each SPSA optimization step, leading to highly noisy and variable estimated SPSA gradients $\hat{g}_i^{spsa}$. This causes an unstable optimization path, requiring more optimization steps for effective convergence. In our task, more optimization steps correspond to more expensive API calls to the LLM, significantly increasing training time and cost. Moreover, unstable optimization makes it difficult to achieve optimal results, resulting in poor model performance.

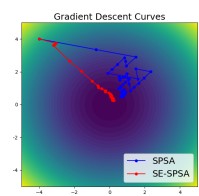

Figure 2: Gradient descent comparison of SPSA and SE-SPSA.

To mitigate this issue, we propose an SPSA-specific variance reduction technique to constrain such stochasticity (i.e., fluctuation amplitude) of the SPSA gradients, enabling faster and more robust convergence. Inspired by baseline-based variance reduction methods (Wu et al., 2018), which theoretically and empirically show that subtracting a suitable constant (termed baseline) can regularize the gradient amplitude to stabilize training, we introduce an SPSA-specific baseline to reduce the variance of SPSA gradients for more stable and accelerated model optimization (See Fig. 2). Formally, we subtract an SPSA-specific baseline value $b_i \in \mathbb{R}$ from the original estimated gradient to form a variance-reduced SPSA gradient estimation $\hat{g}_i^{vr\_spsa}$ as follows:

$$\hat{g}_i^{vr\_spsa}(\phi_i) = \frac{1}{J} \sum_{j=1}^{J} \frac{1}{\mathbf{u}_i^{(j)}} \left( \frac{\mathcal{R}(\phi_i - c_i \mathbf{u}_i^{(j)}) - \mathcal{R}(\phi_i + c_i \mathbf{u}_i^{(j)})}{2c_i} - b_i \right). \tag{7}$$

However, it is a non-trivial challenge to obtain the optimal baseline value $b_i^*$ in Eq. 7. To solve this challenge, in our task, we minimize the variance $\mathrm{Var}(\cdot)$ of $\hat{g}_i^{vr\_spsa}$ to derive the closed-form solution for the optimal baseline $b_i^*$ through our extensive mathematical analysis, as detailed in Theorem 1.

**Theorem 1.** *For the baseline-based SPSA gradient estimation in Eq. 7, the optimal baseline $b_i^*$ minimizing the gradient variance has the closed-form solution ($\mathbb{E}[\cdot]$ denotes the expectation):*

$$b_i^* = \frac{\mathbb{E}_{\mathbf{u}_i} \left[ \frac{1}{\mathbf{u}_i^\top \mathbf{u}_i} \left( R(\phi_i - c_i \mathbf{u}_i) - R(\phi_i + c_i \mathbf{u}_i) \right) \right]}{2c_i \mathbb{E}_{\mathbf{u}_i} \left[ \frac{1}{\mathbf{u}_i^\top \mathbf{u}_i} \right]}, \tag{8}$$

where $\mathbf{u}_i = [u_{i,1}, \cdots, u_{i,M}]$ and $u_{i,m} \sim 0.5 \cdot U(0.5, 1) + 0.5 \cdot U(-1, -0.5)$.

*Proof.* We first derive the variance of the baseline-based gradient estimation in Eq. 7:

$$\mathrm{Var}(\hat{g}_i^{vr\_spsa}) = \mathrm{Var}\left( \frac{1}{J} \sum_{j=1}^{J} \frac{1}{\mathbf{u}_i^{(j)}} \left( \frac{\mathcal{R}(\phi_i - c_i \mathbf{u}_i^{(j)}) - \mathcal{R}(\phi_i + c_i \mathbf{u}_i^{(j)})}{2c_i} - b_i \right) \right)$$

$$= \frac{1}{J} \left( \frac{1}{4c_i^2} \mathbb{E}_{\mathbf{u}_i} \left[ \frac{1}{\mathbf{u}_i^\top \mathbf{u}_i} \left( R(\phi_i - c_i \mathbf{u}_i) - R(\phi_i + c_i \mathbf{u}_i) \right)^2 \right] + b_i^2 \mathbb{E}_{\mathbf{u}_i} \left[ \frac{1}{\mathbf{u}_i^\top \mathbf{u}_i} \right] \right.$$

$$- \frac{b_i}{c_i} \mathbb{E}_{\mathbf{u}_i} \left[ \frac{1}{\mathbf{u}_i^\top \mathbf{u}_i} \left( R(\phi_i - c_i \mathbf{u}_i) - R(\phi_i + c_i \mathbf{u}_i) \right) \right] + b_i^2 \mathbb{E}_{\mathbf{u}_i} \left[ \frac{1}{\mathbf{u}_i} \right]^\top \mathbb{E}_{\mathbf{u}_i} \left[ \frac{1}{\mathbf{u}_i} \right] \tag{9}$$

$$+ \mathbb{E}_{\mathbf{u}_i} \left[ \frac{R(\phi_i - c_i \mathbf{u}_i) - R(\phi_i + c_i \mathbf{u}_i)}{2c_i \mathbf{u}_i} \right]^\top \mathbb{E}_{\mathbf{u}_i} \left[ \frac{R(\phi_i - c_i \mathbf{u}_i) - R(\phi_i + c_i \mathbf{u}_i)}{2c_i \mathbf{u}_i} \right]$$

$$\left. - 2b \mathbb{E}_{\mathbf{u}_i} \left[ \frac{1}{\mathbf{u}_i} \right]^\top \mathbb{E}_{\mathbf{u}_i} \left[ \frac{R(\phi_i - c_i \mathbf{u}_i) - R(\phi_i + c_i \mathbf{u}_i)}{2c_i \mathbf{u}_i} \right] \right).$$

To minimize the variance of $\hat{g}_i^{vr\_spsa}$, we set the derivative of the variance with respect to $b_i$ to zero. Given $\mathbb{E}_{\mathbf{u}_i} \left[ \frac{1}{\mathbf{u}_i} \right] = 0$ (see Lemma 1 in Apps.), the process is formulated as:

$$\frac{\partial}{\partial b_i} [\mathrm{Var}(\hat{g}_i^{vr\_spsa})] = -\frac{1}{Jc_i} \mathbb{E}_{\mathbf{u}_i} \left[ \frac{1}{\mathbf{u}_i^\top \mathbf{u}_i} \left( R(\phi_i - c_i \mathbf{u}_i) - R(\phi_i + c_i \mathbf{u}_i) \right) \right] + \frac{2}{J} \mathbb{E}_{\mathbf{u}_i} \left[ \frac{1}{\mathbf{u}_i^\top \mathbf{u}_i} \right] b_i = 0. \tag{10}$$

$\square$

Table 1: Quantitative comparisons on the SST-2 (Wang et al., 2018), QNLI (Wang et al., 2018) and Medical Q/A (Han et al., 2023) datasets.

| Models | SST-2 | | | | QNLI | | | | Medical Q/A | | | |
|---|---|---|---|---|---|---|---|---|---|---|---|---|
| | $P_{GS}$ | $P_{I\text{-}LS}$ | $P_{O\text{-}LS}$ | $U_{ACC}$ | $P_{GS}$ | $P_{I\text{-}LS}$ | $P_{O\text{-}LS}$ | $U_{ACC}$ | $P_{GS}$ | $P_{I\text{-}LS}$ | $P_{O\text{-}LS}$ | $U_{Rouge_{1/2/L}}$ |
| PlainText | 0.382 | 0.000 | 0.041 | 0.959 | 0.257 | 0.000 | 0.081 | 0.919 | 0.550 | 0.000 | 0.628 | 0.247 / 0.060 / 0.218 |
| SanText(Yue et al., 2021) | 0.657 | 0.836 | 0.463 | 0.537 | 0.668 | 0.658 | 0.505 | 0.495 | 0.853 | 0.664 | 0.817 | 0.130 / 0.014 / 0.113 |
| SanText+(Yue et al., 2021) | 0.566 | 0.435 | 0.358 | 0.642 | 0.468 | 0.272 | 0.503 | 0.497 | 0.697 | 0.347 | 0.727 | 0.178 / 0.030 / 0.153 |
| CusText(Chen et al., 2023a) | 0.577 | 0.694 | 0.390 | 0.610 | 0.469 | 0.262 | 0.463 | 0.537 | 0.720 | 0.343 | 0.672 | 0.200 / 0.038 / 0.178 |
| CusText+(Chen et al., 2023a) | 0.571 | 0.433 | 0.242 | 0.758 | 0.418 | 0.196 | 0.372 | 0.628 | 0.640 | 0.116 | 0.671 | 0.201 / 0.043 / 0.173 |
| HaS(Chen et al., 2023b) | 0.536 | 0.479 | 0.137 | 0.863 | 0.327 | 0.142 | 0.316 | 0.684 | 0.563 | 0.423 | 0.717 | 0.177 / 0.025 / 0.151 |
| LeQP | 0.769 | 0.638 | 0.247 | 0.753 | 0.813 | 0.672 | 0.486 | 0.514 | 0.740 | 0.513 | 0.758 | 0.166 / 0.025 / 0.114 |
| PrivateChat(*Caesar*) | 0.825 | 0.857 | **0.999** | 0.864 | 0.860 | **0.800** | 0.937 | 0.712 | 0.864 | 0.767 | **0.982** | **0.232 / 0.045 / 0.211** |
| PrivateChat(*DES*) | 0.837 | 0.834 | 0.973 | 0.856 | 0.875 | 0.759 | 0.949 | 0.804 | **0.952** | 0.714 | 0.943 | 0.182 / 0.040 / 0.151 |
| PrivateChat(*AES*) | **0.845** | **0.889** | 0.982 | **0.901** | 0.835 | 0.746 | **0.960** | 0.813 | 0.948 | **0.857** | 0.974 | 0.216 / 0.043 / 0.181 |
| PrivateChat(*ChaCha20*) | 0.833 | 0.842 | 0.975 | 0.874 | **0.907** | 0.714 | 0.917 | 0.796 | 0.946 | 0.715 | 0.972 | 0.191/ 0.042 / 0.179 |

Finally, by solving Eq. 10, we derive the optimal baseline $b_i^*$ in Eq. 8 (refer to Apps. for detailed derivations). Given that the expected values in Eq. 8 are intractable due to the continuity of $\mathbf{u}_i$, we exploit the sample mean to estimate $b_i^*$ as follows:

$$\hat{b}_i^* = \frac{\sum_{j=1}^{J} \frac{1}{\mathbf{u}_i^{(j)\top}\mathbf{u}_i^{(j)}} \left( R(\phi_i - c_i\mathbf{u}_i^{(j)}) - R(\mathbf{u}_i^{(j)}) \right)}{2c_i \sum_{j=1}^{J} \frac{1}{\mathbf{u}_i^{(j)\top}\mathbf{u}_i^{(j)}}}. \tag{11}$$

where $\{\mathbf{u}_i^{(j)}\}_{j=1}^{J}$ are randomly sampled perturbation vectors. Having derived the optimal baseline $\hat{b}_i^*$ via Eq. 11 and substituting it back into Eq. 7, we develop a new variant of SPSA, SE-SPSA, which provides more stable gradient estimation, better approximating the correct gradient direction for more reliable convergence. In our task, this also means fewer API calls to LLMs and more effective prompt generation, thus enabling economical and efficient private conversations with cloud LLMs.

## 4 EXPERIMENTS

**Tasks.** Our study focuses on sentiment classification and question-answering (Q/A) tasks. Following (Yue et al., 2021; Chen et al., 2023a), we evaluate our approach on the SST-2 and QNLI classification datasets from the GLUE benchmark (Wang et al., 2018), containing over 1.8k and 5.2k test samples, respectively. To simulate interactions between users and LLMs, we further evaluate our method on the medical Q/A dataset, which contains 100 real-world Q/A pairs from a collaborative medical platform (Han et al., 2023).

**Setup.** In our system prompt perturbation module, we randomly generate $R = 50$ codes to form a codebook, each code consisting of $N_c = 1$ ASCII character. The perturbation threshold $\varepsilon$ is set to 0.7. For the black-box optimization, the optimization steps $I$ is set to 8 and the number of sampled perturbation vectors $J$ is set to 5. Following (Oh et al., 2023), both the perturbation coefficient $c_i$ and the learning rate $a_i$ are dynamically adjustable. Considering the widespread use of GPT-4 (OpenAI, 2023b), we select it as the cloud LLM for training and evaluation. During the training phase, we use 5 samples from the SST-2 training dataset (Wang et al., 2018) for prompt optimization.

**Comparison Methods.** We compare our PrivateChat with two main types of privacy-preserving methods: (i) **Local Differential privacy (LDP) methods** (*SanText* (Yue et al., 2021) and *CusText* (Chen et al., 2023a)), which enhance privacy by adding noise to input data. (ii) **Anonymization method** (i.e., *HaS* (Chen et al., 2023b)), which employs a local LLM to replace privacy entities (e.g., names, numbers, and locations) with synonyms. Notably, LDP requires model fine-tuning to maintain utility, while anonymization methods focus solely on masking private entities. As a result, neither approach is well-suited for comprehensive protection in our daily chat scenarios. However, since these methods are not dependent on specific LLM architectures, they can be adapted to our

Table 2: Results on different private system prompts.

| Models | $P_{GS}$ | $P_{I\text{-}LS}$ | $P_{O\text{-}LS}$ | $U_{ACC}$ |
|---|---|---|---|---|
| DP-based Prompt | 0.756 | 0.654 | 0.540 | 0.455 |
| Anon-based Prompt | 0.736 | 0.250 | 0.860 | 0.596 |
| Token-level | 0.667 | 0.571 | 0.950 | 0.769 |
| Word-level | 0.794 | 0.714 | 0.802 | 0.657 |
| Character-level (Ours) | **0.825** | **0.857** | **0.999** | **0.864** |

Figure 3: Privacy & utility performance with different perturbation threshold $\varepsilon$.

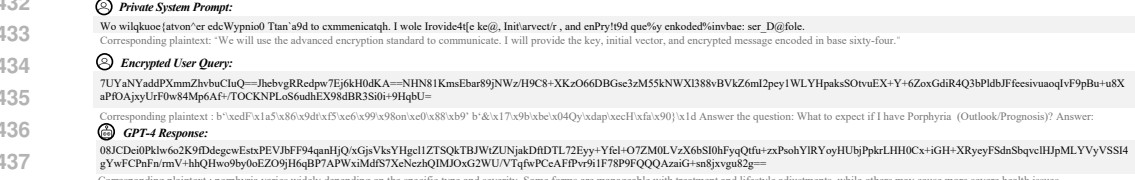

☺ *Private System Prompt:*
Wo wilqkuoe{atvon^er edcWypnio0 Ttan'a9d to cxmmenicatqh. I wole Irovide4t[e ke@, Init\arvect/r , and enPry!t9d que%y enkoded%invbae: ser_D@fole.
Corresponding plaintext: "We will use the advanced encryption standard to communicate. I will provide the key, initial vector, and encrypted message encoded in base sixty-four."

☺ *Encrypted User Query:*
7UYaNYaddPXmmZhvbuCIuQ==JhebvgRRedpw7Ej6kH0dKA==NHN81KmsEbar89jNWz/H9C8+XKzO66DBGse3zM55kNWXl388vBVkZ6mI2pey1WLYHpaksSOtvuEX+Y+6ZoxGdiR4Q3bPldbJFfeesivuaoqlvF9pBu+u8X aPfOAjxyUrF0w84Mp6AF+/TOCKNPLoS6udhEX98dBR3Si0i+9HqbU=
Corresponding plaintext : b'\xedF\x1a5\x86\x9df\xf5\xe6\x99\x98on\xe0\x88\xb9' b'&\x17\x9b\xbe\x04Qy\xdap\xecH\xfa\x90}'\x1d Answer the question: What to expect if I have Porphyria (Outlook/Prognosis)? Answer:

☺ *GPT-4 Response:*
08JCDei0Pklw6o2K9fDdegcwEstxPEVJbFF94qanHjQ/xGjsVksYHgcl1ZTSQkTBJWtZUNjakDftDTL72Eyy+Yfel+O7ZM0LVzX6bSI0hFyqQtfu+zxPsohYlRYoyHUbjPpkrLHH0Cx+iGH+XRyeyFSdnSbqvclHJpMLYVyVSSI4 gYwFCPnFn/rmV+hhQHwo9by0oEZO9jH6qBP7APWxiMdfS7XeNezhQIMJOxG2WU/VTqfwPCeAFfPvr9i1F78P9FQQQAzaiG+sn8jxvgu82g==
Corresponding plaintext : porphyria varies widely depending on the specific type and severity. Some forms are manageable with treatment and lifestyle adjustments, while others may cause more severe health issues.

Figure 4: An encrypted medical Q/A example with GPT-4 (OpenAI, 2023b) under PrivateChat.

setting. Additionally, we design another baseline for comparison: **(iii) Learnable query perturbation (LeQP)**, that maps plaintext user queries into perturbed text with a learnable perturbation model. The model is trained with our SE-SPSA optimizer, using 200 samples from the SST-2 training dataset (Wang et al., 2018). Unlike our PrivateChat, which employs encryption algorithms to protect user queries while perturbing the system prompt, LeQP adaptively perturbs user queries without an additional system prompt.

**Evaluation on Classification Tasks.** Following (Yue et al., 2021; Chen et al., 2023a; Tong et al., 2023), we use two widely-used metrics to evaluate **privacy protection levels** by measuring model's robustness against common attacks: (1) Local Semantic Protection Degree ($P_{LS}$), which exploits the *embedding inversion attack* (Qu et al., 2021) to measure the local, token-wise semantic privacy level by comparing the semantic embedding similarity between the private token and plaintext token ($P_{I\_LS}$ and $P_{O\_LS}$ denote the local semantic protection degree of the perturbed LLM inputs and that of the LLM outputs, respectively). (2) Global Semantic Protection Degree ($P_{GS}$), which adopts the *input inference attack* (Yue et al., 2021) to measure the global semantic privacy level of the perturbed LLM inputs by computing the rate of incorrect inference on partially masked tokens. Following (Yue et al., 2021; Chen et al., 2023a), we measure the **utility level** by the accuracy ($U_{ACC}$) of LLM responses. As shown in Table 1, the DP methods (Yue et al., 2021; Chen et al., 2023a) and the learnable perturbation method (LeQP) make the input text incoherent, significantly reducing LLM comprehension and response accuracy. The anonymization method (Chen et al., 2023b) fails to fully conceal sensitive information, resulting in poor privacy-preserving performance. In contrast, PrivateChat excels across all privacy and utility metrics and achieves comparable utility to the plaintext method (i.e., plaintext user input). This superior performance is attributed to: **(i)** Customized encryption and the private system prompt ensure secure communications that are only interpreted by the user and the LLM. **(ii)** The black-box optimizer enables the generated system prompt to effectively guide the LLM to produce encrypted and accurate responses.

**Evaluation on Question-answering (Q/A) Task.** To simulate daily interactions between users and LLMs, we evaluate our method on the medical Q/A dataset. For **privacy protection level** assessment, we use the Local Semantic Protection Degree ($P_{LS}$) and Global Semantic Protection Degree ($P_{GS}$) mentioned above. Following (Xiao et al., 2023), we assess the **utility level** using three Rouge criteria: $U_{Rouge_1}$, $U_{Rouge_2}$ and $U_{Rouge_L}$. As shown in Tab. 1, PrivateChat outperforms other methods in both privacy and utility levels. We show a Q/A chat example in Fig. 4, demonstrating that our method enables secure, effective communication between the user and the LLM.

**Ablation Study on System Prompt Perturbation Model.** Our system prompt perturbation model is designed to generate effective private system prompts. We first show its effectiveness by comparing our prompt with those generated by the differential privacy method (DP-based Prompt) (Chen et al., 2023a) and anonymization method (Anon-based Prompt) (Vats et al., 2023) on the SST-2 dataset (Wang et al., 2018). The DP-based Prompt incorporates random noise into the plaintext prompt, while the Anon-based Prompt replaces encryption details with synonyms. Additionally, we evaluate our character-level perturbation strat-

Table 3: Comparison of different black-box optimizers.

| Models | $P_{GS}$ | $P_{I\_LS}$ | $P_{O\_LS}$ | $U_{ACC}$ | Training time | No. of API Calls |
|---|---|---|---|---|---|---|
| Random Search | 0.815 | 0.667 | 0.854 | 0.498 | 4837s | 1100 |
| DDPG | 0.803 | 0.750 | 0.945 | 0.770 | 5452s | 1039 |
| BAR | 0.812 | 0.714 | 0.895 | 0.668 | 4176s | 970 |
| BlackVIP | 0.813 | **0.857** | 0.931 | 0.783 | 1897s | 440 |
| SPSA | 0.808 | 0.833 | 0.870 | 0.739 | 2583s | 590 |
| SE-SPSA | **0.825** | **0.857** | **0.999** | **0.864** | **345**s | **80** |

Table 4: Results on various cloud LLMs.

| Models | $P_{GS}$ | $P_{I\_LS}$ | GPT-4V | | Sonnet | | Opus | |
|---|---|---|---|---|---|---|---|---|
| | | | $P_{O\_LS}$ | $U_{ACC}$ | $P_{O\_LS}$ | $U_{ACC}$ | $P_{O\_LS}$ | $U_{ACC}$ |
| SanText | 0.657 | 0.836 | 0.258 | 0.742 | 0.613 | 0.387 | 0.380 | 0.620 |
| CusText+ | 0.571 | 0.433 | 0.231 | 0.769 | 0.377 | 0.623 | 0.446 | 0.554 |
| HaS | 0.536 | 0.479 | 0.127 | 0.873 | 0.277 | 0.723 | 0.183 | 0.817 |
| PrivateChat | **0.825** | **0.857** | **0.990** | **0.891** | **0.990** | **0.730** | **0.920** | **0.836** |

egy against word-level and token-level ones. As shown in Tab. 2, our optimization-based method, enhanced by feedback from LLMs, outperforms both DP-based and Anon-based methods. Compared to word-level and token-level perturbations, character-level perturbation offers higher robustness, achieving better performance. Moreover, we assess the impact of the perturbation threshold $\varepsilon$

in our system prompt perturbation model. Fig. 3 shows PrivateChat's performance under varying $\varepsilon$ settings, where $\varepsilon = 0$ means all characters are perturbed. The privacy metric is the average of $P_{GS}$ and $P_{LS}$. It is clear that as $\varepsilon$ increases, utility improves but privacy decreases, and setting $\varepsilon = 0.7$ offers optimal overall performance.

**Comparison of Optimization Methods.** Our SE-SPSA is designed for effective and economical black-box optimization. To assess its effectiveness, we compare it with the original SPSA (Spall, 1992a). As shown in Fig. 5, benefiting from our baseline-based variance reduction strategy, SE-SPSA achieves more stable and accelerated convergence than the original SPSA. Furthermore, we compare our method with other black-box optimizers, such as random search (Bergstra & Bengio, 2012), DDPG (Lillicrap et al., 2015), BAR (Tsai et al., 2020) and BlackVIP (Oh et al., 2023) on the SST-2 dataset (Wang et al., 2018). As shown in Tab. 3, leveraging our variance reduction strategy, SE-SPSA significantly cuts training time and costs while achieving the best performance.

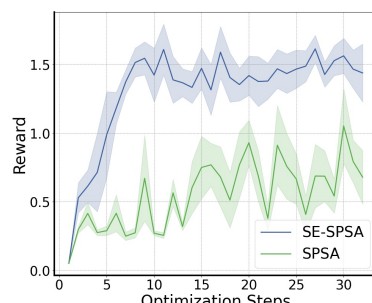

Figure 5: Comparison of reward curves among SPSA and SE-SPSA.

**Experiments on Various Cloud LLMs.** To demonstrate the generality of our framework, we assess its performance using popular cloud LLMs other than GPT-4 (OpenAI, 2023b), including GPT-4V (OpenAI, 2023b), Claude3 Sonnet (Anthropic, 2023), and Claude3 Opus (Anthropic, 2023) on the SST-2 dataset (Wang et al., 2018). As shown in Tab. 4, our PrivateChat exhibits impressive classification accuracy across various cloud LLMs.

## 5 DISCUSSION

Here, we note that compared with other privacy-preserving methods, our PrivateChat has significant differences and benefits as follows: **1) Black-box Adaptability**: Traditional privacy-preserving methods, such as homomorphic encryption and federated learning, are generally limited to service providers and inaccessible to clients without access to model parameters. In contrast, our approach does not rely on access to model parameters or architectures, making it more adaptable for real-world black-box scenarios. **2) Utility-Privacy Trade-off**: Although local differential privacy (LDP) can sanitize user queries locally, it often leads to unacceptable utility loss when a high degree of privacy is necessary. Our method addresses this trade-off between privacy protection and utility with a novel encryption framework. **3) Innovation and Inspiration**: Our work serves as an exploratory and foundational contribution to the field of LLM privacy protection. We are the first to propose an encryption framework designed for secure communication with black-box LLMs, with the potential to significantly influence future research and applications in this area.

## 6 CONCLUSION

In this paper, we have proposed PrivateChat, a novel private communication framework for encrypted interactions between users and cloud black-box LLMs. Our PrivateChat consists of three main modules: a client-end encryption module that encrypts user queries with the user-customized method and key, a system prompt perturbation module that safely instructs the LLM to process encrypted user queries and produce encrypted responses, and a client-end decryption module that converts the encrypted LLM responses back into plaintext. To optimize our framework, we have designed SE-SPSA, an enhanced black-box optimizer that significantly reduces the training time and costs, and improves the performance of the original SPSA via our baseline-based variance reduction strategy.

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
