# PrivateChat: A Secure Encrypted Communication Framework with Black-box LLMs (Technical Appendices)

In this appendix, we first provide additional details of our method in Section 1. Then, in Section 2, we describe more experimental details. Finally, in Section 3, we present additional experiments, including a user study, various ablation studies, and case studies of our PrivateChat.

## 1 More Details of PrivateChat

Here, we present the details of how we build the client-end encryption module, the client-end decryption module, the system prompt perturbation module and the sample-efficient black-box optimization framework.

### 1.1 Client-end Encryption Module and Client-end Decryption Module

In our client-end encryption module, we employ various encryption algorithms (e.g., Caesar, AES, DES, and ChaCha20) to convert the user's plaintext queries into ciphertext. In our client-end decryption module, we utilize the corresponding decryption algorithms to transform the encrypted responses back into plaintext. Next, we elaborate on the encryption algorithms used in this paper.

**Caesar cipher** is a traditional encryption method where each letter in the plaintext is shifted a certain number of positions up or down the alphabet. This substitution is consistent throughout the entire message. The key to the Caesar cipher is the number of positions each letter in the plaintext is moved. For example, if the key is 3, the letter 'A' in the plaintext will be replaced by 'D'.

**AES (Advanced Encryption Standard) cipher** is a widely used symmetric encryption algorithm, known for its efficiency and robustness in securing electronic data. The core operations of AES include substitution, shifting rows, mixing columns, and adding a round key (XORing the block with a key derived from the original key). AES supports multiple key lengths, and the same key is used for both encryption and decryption.

**DES (Data Encryption Standard) cipher** is a symmetric encryption algorithm that was widely used to secure electronic data. During each round of DES, the data blocks are divided into two halves. The right half undergoes a complex function that involves expansion, substitution, and permutation, and then it is combined with the left half using an XOR operation. The decryption process involves applying these steps in reverse order, using the round keys in reverse.

**ChaCha20 cipher** is a modern stream cipher known for its high performance and security. Unlike block ciphers like AES and DES that process data in blocks, ChaCha20 operates as a stream cipher. It generates a long keystream of pseudo-random bits, which is then XORed with the plaintext to produce ciphertext. In addition to the key, ChaCha20 also uses a nonce and a counter. As with other symmetric ciphers, the same key is used for both encryption and decryption.

### 1.2 System Prompt Perturbation Module

We design a learnable system prompt perturbation model that adaptively perturbs the initial plaintext prompt $\Pi$ to generate a private system prompt $\tilde{\Pi}$. As illustrated in Fig. 1, take the word 'Caesar' as an example. Specifically, **(i)** $P^E = \{p_n^E\}_{n=1}^6$ represents the perturbation probability distribution, where each $p_n^E$ denotes the probability of perturbing the $n^{th}$ character $\pi_n$ in the word 'Caesar'. Once this probability $p_n^E$ exceeds the perturbation threshold $\varepsilon$, the character $\pi_n$ is replaced with a corresponding code from the codebook. In this case, characters 'C', 'e', 's', and 'r' require perturbation.

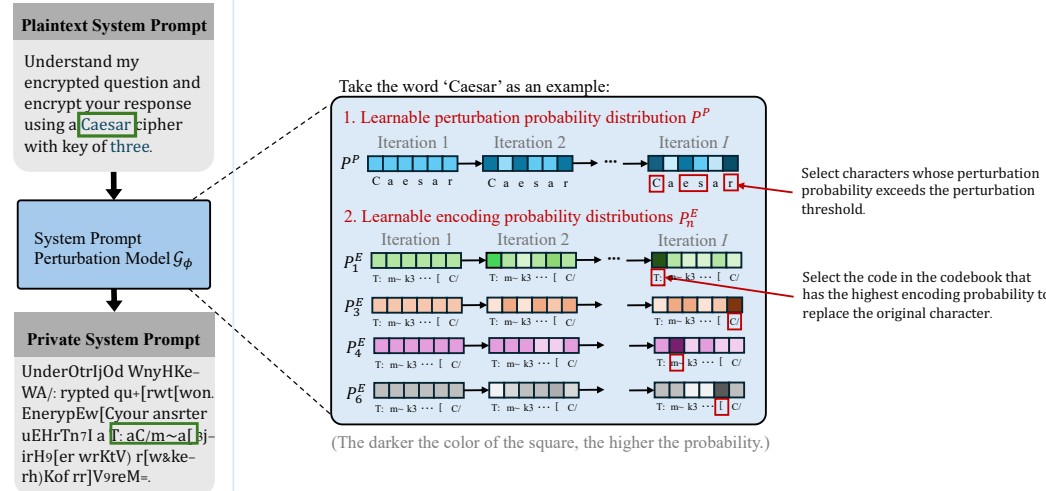

Figure 1: The pipeline of our system prompt perturbation module.

**(ii)** $P_n^E = \{p_{n,r}^E\}_{r=1}^R$ is the encoding probability distribution, where $p_{n,r}^E$ denotes the probability that the character $\pi_n$ should be perturbed as $\mathcal{C}_r$ ($\mathcal{C}_r$ denotes the $r^{th}$ code within a codebook containing a total of $R$ codes). Here, each code in the codebook is a random combination of $N_c = 2$ ASCII characters. For instance, for the original character 'C', the code 'T:' within its corresponding codebook has the highest encoding probability, so 'C' is replaced with 'T:'. Consequently, using this system prompt perturbation model, we can perturb the word 'Caesar' in the plaintext system prompt to generate the word 'T:aC/m a[ ' in the private system prompt.

### 1.3 SAMPLE-EFFICIENT BLACK-BOX OPTIMIZATION FRAMEWORK

To enable efficient and economical black-box optimization, we introduce a baseline-based variance reduction strategy specific to Simultaneous Perturbation Stochastic Approximation (SPSA). This strategy stabilizes and accelerates the convergence of the original SPSA, enhancing model performance while reducing training time and costs. Specifically, we subtract a baseline value $b_i \in \mathbb{R}$ from the original SPSA gradient estimate to form a variance-reduced gradient estimation $\hat{g}_i^{vr\_spsa}$ as follows:

$$\hat{g}_i^{vr\_spsa}(\phi_i) = \frac{1}{J}\sum_{j=1}^{J}\frac{1}{\mathbf{u}_i^{(j)}}\left(\frac{\mathcal{R}(\phi_i - c_i\mathbf{u}_i^{(j)}) - \mathcal{R}(\phi_i + c_i\mathbf{u}_i^{(j)})}{2c_i} - b_i\right). \tag{1}$$

Lemma 1 below theoretically proves that our baseline-based gradient estimation is unbiased relative to the original gradient estimation, ensuring that while the variance of the gradient estimates is reduced, their accuracy in indicating the correct direction for steepest reward ascent is preserved.

**Lemma 1.** *The baseline-based gradient estimation is unbiased to the original one:* $\mathbb{E}\left[\hat{g}_i^{vr\_spsa}(\phi_i)\right] = g_i^{spsa}(\phi_i).$

*Proof.* We first rewrite our baseline-based gradient $\hat{g}_i^{vr\_spsa}$ as follows:

$$\mathbb{E}\left[\hat{g}_i^{vr\_spsa}(\phi_i)\right] = \mathbb{E}\left[\frac{1}{J}\sum_{j=1}^{J}\frac{1}{\mathbf{u}_i^{(j)}}\left(\frac{\mathcal{R}(\phi_i - c_i\mathbf{u}_i^{(j)}) - \mathcal{R}(\phi_i + c_i\mathbf{u}_i^{(j)})}{2c_i} - b_i\right)\right]$$
$$= \mathbb{E}_{\mathbf{u}_i}\left[\frac{1}{\mathbf{u}_i}\left(\frac{R(\phi_i - c_i\mathbf{u}_i) - R(\phi_i + c_i\mathbf{u}_i)}{2c_i}\right)\right] - b_i\mathbb{E}_{\mathbf{u}_i}\left[\frac{1}{\mathbf{u}_i}\right] \tag{2}$$
$$= g_i^{spsa}(\phi_i) - b_i\mathbb{E}_{\mathbf{u}_i}\left[\frac{1}{\mathbf{u}_i}\right].$$

Since $u_{i,m} \sim 0.5 \cdot U(0.5, 1) + 0.5 \cdot U(-1, -0.5)$, we compute the expected value of $\frac{1}{u_{i,m}}$ as follows:

$$\mathbb{E}_{u_{i,m}}\left[\frac{1}{u_{i,m}}\right] = \int_{-1}^{-0.5}\frac{1}{u_{i,m}}\cdot\frac{1}{-0.5+1}du_{i,m} + \int_{0.5}^{1}\frac{1}{u_{i,m}}\cdot\frac{1}{1-0.5}du_{i,m} = 0. \tag{3}$$

Based on the derivation above, we can obtain $\mathbb{E}_{\mathbf{u}_i}\left[\frac{1}{\mathbf{u}_i}\right] = \mathbf{0}$ and thus $\mathbb{E}\left[\hat{g}_i^{vr\text{-}spsa}(\phi_i)\right] = g_i^{spsa}(\phi_i)$.

$\square$

Additionally, we minimize the variance $\text{Var}(\cdot)$ of $\hat{g}_i^{vr\text{-}spsa}$ to derive the closed-form solution for the optimal baseline $b_i^*$ in Eq. 4 through our extensive mathematical analysis. Here, we provide a detailed derivation of our optimal baseline $b_i^*$.

$$b_i^* = \frac{\mathbb{E}_{\mathbf{u}_i}\left[\frac{1}{\mathbf{u}_i^\top \mathbf{u}_i}\left(R(\phi_i - c_i\mathbf{u}_i) - R(\phi_i + c_i\mathbf{u}_i)\right)\right]}{2c_i\mathbb{E}_{\mathbf{u}_i}\left[\frac{1}{\mathbf{u}_i^\top \mathbf{u}_i}\right]}, \tag{4}$$

*Proof.* We first derive the variance of the baseline-based gradient estimation in Eq. 1:

$$
\begin{aligned}
\text{Var}(\hat{g}_i^{vr\text{-}spsa}) &= \text{Var}\left(\frac{1}{J}\sum_{j=1}^{J}\frac{1}{\mathbf{u}_i^{(j)}}\left(\frac{\mathcal{R}(\phi_i - c_i\mathbf{u}_i^{(j)}) - \mathcal{R}(\phi_i + c_i\mathbf{u}_i^{(j)})}{2c_i} - b_i\right)\right) \\
&= \frac{1}{J^2}\text{Var}\left(\sum_{j=1}^{J}\frac{1}{\mathbf{u}_i^{(j)}}\left(\frac{\mathcal{R}(\phi_i - c_i\mathbf{u}_i^{(j)}) - \mathcal{R}(\phi_i + c_i\mathbf{u}_i^{(j)})}{2c_i} - b_i\right)\right) \\
&= \frac{1}{J}\text{Var}\left(\frac{1}{\mathbf{u}_i^{(j)}}\left(\frac{\mathcal{R}(\phi_i - c_i\mathbf{u}_i^{(j)}) - \mathcal{R}(\phi_i + c_i\mathbf{u}_i^{(j)})}{2c_i} - b_i\right)\right) \\
&= \frac{1}{J}\text{Var}\left(\underbrace{\frac{1}{\mathbf{u}_i}\left(\frac{\mathcal{R}(\phi_i - c_i\mathbf{u}_i) - \mathcal{R}(\phi_i + c_i\mathbf{u}_i)}{2c_i} - b_i\right)}_{g_{\mathbf{u}_i}}\right) \\
&= \frac{1}{J}\left(\mathbb{E}_{\mathbf{u}_i}\left[\left(g_{\mathbf{u}_i} - \mathbb{E}_{\mathbf{u}_i}[g_{\mathbf{u}_i}]\right)^\top\left(g_{\mathbf{u}_i} - \mathbb{E}_{\mathbf{u}_i}[g_{\mathbf{u}_i}]\right)\right]\right) \\
&= \frac{1}{J}\left(\mathbb{E}_{\mathbf{u}_i}[g_{\mathbf{u}_i}^\top g_{\mathbf{u}_i}] - \mathbb{E}_{\mathbf{u}_i}[g_{\mathbf{u}_i}]^\top \mathbb{E}_{\mathbf{u}_i}[g_{\mathbf{u}_i}]\right)
\end{aligned}
\tag{5}
$$

$$
\begin{aligned}
\mathbb{E}_{\mathbf{u}_i}[g_{\mathbf{u}_i}^\top g_{\mathbf{u}_i}] &= \mathbb{E}_{\mathbf{u}_i}\left[\frac{1}{\mathbf{u}_i^\top \mathbf{u}_i}\left(\frac{(R(\phi_i - c_i\mathbf{u}_i) - R(\phi_i + c_i\mathbf{u}_i))}{2c_i} - b_i\right)^2\right] \\
&= \frac{1}{4c_i^2}\mathbb{E}_{\mathbf{u}_i}\left[\frac{1}{\mathbf{u}_i^\top \mathbf{u}_i}\left(R(\phi_i - c_i\mathbf{u}_i) - R(\phi_i + c_i\mathbf{u}_i)\right)^2\right] + b_i^2\mathbb{E}_{\mathbf{u}_i}\left[\frac{1}{\mathbf{u}_i^\top \mathbf{u}_i}\right] \\
&\quad - \frac{b_i}{c_i}\mathbb{E}_{\mathbf{u}_i}\left[\frac{1}{\mathbf{u}_i^\top \mathbf{u}_i}\left(R(\phi_i - c_i\mathbf{u}_i) - R(\phi_i + c_i\mathbf{u}_i)\right)\right]
\end{aligned}
\tag{6}
$$

$$
\begin{aligned}
\mathbb{E}_{\mathbf{u}_i}[g_{\mathbf{u}_i}]^\top \mathbb{E}_{\mathbf{u}_i}[g_{\mathbf{u}_i}] &= \mathbb{E}_{\mathbf{u}_i}\left[\frac{R(\phi_i - c_i\mathbf{u}_i) - R(\phi_i + c_i\mathbf{u}_i)}{2c_i\mathbf{u}_i} - \frac{b_i}{\mathbf{u}_i}\right]^\top \mathbb{E}_{\mathbf{u}_i}\left[\frac{R(\phi_i - c_i\mathbf{u}_i) - R(\phi_i + c_i\mathbf{u}_i)}{2c_i\mathbf{u}_i} - \frac{b_i}{\mathbf{u}_i}\right] \\
&= \mathbb{E}_{\mathbf{u}_i}\left[\frac{R(\phi_i - c_i\mathbf{u}_i) - R(\phi_i + c_i\mathbf{u}_i)}{2c_i\mathbf{u}_i}\right]^\top \mathbb{E}_{\mathbf{u}_i}\left[\frac{R(\phi_i - c_i\mathbf{u}_i) - R(\phi_i + c_i\mathbf{u}_i)}{2c_i\mathbf{u}_i}\right] \\
&\quad - 2b_i\mathbb{E}_{\mathbf{u}_i}\left[\frac{1}{\mathbf{u}_i}\right]^\top \mathbb{E}_{\mathbf{u}_i}\left[\frac{R(\phi_i - c_i\mathbf{u}_i) - R(\phi_i + c_i\mathbf{u}_i)}{2c_i\mathbf{u}_i}\right] + b_i^2\mathbb{E}_{\mathbf{u}_i}\left[\frac{1}{\mathbf{u}_i}\right]^\top \mathbb{E}_{\mathbf{u}_i}\left[\frac{1}{\mathbf{u}_i}\right]
\end{aligned}
\tag{7}
$$

To minimize the variance of $\hat{g}_i^{vr\text{-}spsa}$, we set the derivative of the variance with respect to $b_i$ to zero. Given $\mathbb{E}_{\mathbf{u}_i}\left[\frac{1}{\mathbf{u}_i}\right] = 0$ (see Lemma 1), the process is formulated as:

$$\frac{\partial}{\partial b_i}[\text{Var}(\hat{g}_i^{vr\text{-}spsa})] = -\frac{1}{Jc_i}\mathbb{E}_{\mathbf{u}_i}\left[\frac{1}{\mathbf{u}_i^\top \mathbf{u}_i}\left(R(\phi_i - c_i\mathbf{u}_i) - R(\phi_i + c_i\mathbf{u}_i)\right)\right] + \frac{2}{J}\mathbb{E}_{\mathbf{u}_i}\left[\frac{1}{\mathbf{u}_i^\top \mathbf{u}_i}\right]b_i = 0 \tag{8}$$

$$\implies b_i^* = \frac{\mathbb{E}_{\mathbf{u}_i}\left[\frac{1}{\mathbf{u}_i^\top \mathbf{u}_i}\left(R(\phi_i - c_i\mathbf{u}_i) - R(\phi_i + c_i\mathbf{u}_i)\right)\right]}{2c_i\mathbb{E}_{\mathbf{u}_i}\left[\frac{1}{\mathbf{u}_i^\top \mathbf{u}_i}\right]}. \tag{9}$$

$\square$

Algorithm 1 summarizes our SE-SPSA algorithm.

---

**Algorithm 1** SE-SPSA Algorithm for Black-Box Optimization

---

**Input:** the total number $I$ of optimization steps, the total number $J$ of sampled perturbation vectors in each optimization step, the objective function $\mathcal{R}(\cdot)$, the scaling parameter $S_a$ and the stabilization parameter $S_o$ for the learning rate $a$, the scaling parameter $S_c$ and the perturbation magnitude $S_p$ for the perturbation coefficient $c$.

Initialize the model parameters $\phi_0$

**for** $i \leftarrow 0$ to $I - 1$ **do**

$\quad a_i = \frac{S_a}{(i+S_o)^{S_a}}$

$\quad c_i = \frac{S_c}{i^{S_p}}$

$\quad$ **for** $t \leftarrow 1$ to $J$ **do**

$\quad\quad$ Sample random perturbation vector $\mathbf{u}_i^{(j)}$

$\quad$ **end for**

$\quad$ // Derive the optimal baseline for variance reduction

$\quad \hat{b}_i^* = \frac{\sum_{j=1}^{J} \frac{1}{\mathbf{u}_i^{(j)\top}\mathbf{u}_i^{(j)}}\left(R(\phi_i - c_i\mathbf{u}_i^{(j)}) - R(\phi_i + c_i\mathbf{u}_i^{(j)})\right)}{2c_i \sum_{j=1}^{J} \frac{1}{\mathbf{u}_i^{(j)\top}\mathbf{u}_i^{(j)}}}$

$\quad \hat{g}_i^{vr\_spsa}(\phi_i) = \frac{1}{J}\sum_{j=1}^{J}\frac{1}{\mathbf{u}_i^{(j)}}\left(\frac{\mathcal{R}(\phi_i - c_i\mathbf{u}_i^{(j)}) - \mathcal{R}(\phi_i + c_i\mathbf{u}_i^{(j)})}{2c_i} - \hat{b}_i^*\right)$

$\quad \phi_{i+1} = \phi_i - a_i\hat{g}_i^{vr\_spsa}(\phi_i)$

**end for**

---

## 2 MORE DETAILS OF EXPERIMENTAL SETUP

### 2.1 IMPLEMENTATION DETAILS

In our black-box optimization module, we set the scaling parameter $S_a = 0.05$ and the stabilization parameter $S_o = 1.0$ to calculate the learning rate $a_i$; we set the scaling parameter $S_c = 0.2$ and the stabilization parameter $S_p = 0.1$ to calculate perturbation coefficient $c_i$ (see Algorithm 1 for the calculation formula). To ensure a fair evaluation, we introduce an additional system prompt for each method, such as "Answer without writing analysis steps", to prevent the exposure of private information (e.g., user queries being repeated) in the analysis steps of LLM responses, thereby improving the accuracy of our privacy assessment.For our method, we include a system prompt like "Please use Python code to answer my question", to reduce randomness in LLM responses.

All experiment are implemented using PyTorch with one RTX 3090 GPU. The learnable query perturbation method we designed for comparison experiments uses the same character-level perturbation strategy as our PrivateChat. This model consists of an embedding layer, an LSTM layer, and a fully connected layer, and is optimized with our SE-SPSA black-box optimizer. The inputs to the network are user queries, while the outputs provide the perturbation probability distribution and encoding probability distribution required by our character-level perturbation strategy.

### 2.2 METRICS

**Local Semantic Protection Degree** ($P_{LS}$). Following (Tong et al., 2023), we adopt the *embedding inversion attack* (Qu et al., 2021) to measure the local, token-wise semantic privacy level of the perturbed LLM inputs and that of the LLM outputs. Specifically, given the embeddings of tokens in the private text, the embedding inversion attack (Yue et al., 2021) aims to find the top 5 nearest neighbors in the embedding space. The corresponding tokens of these nearest neighbors are then used to replace the tokens in the private text, inferring the original text. The success rate of this attack (i.e., the accuracy of token prediction) is denoted as $R_{LS}$, which is inversely proportional to the privacy protection capability: the higher the $R_{LS}$, the lower the privacy protection. Therefore, the privacy protection level is defined as $P_{LS} = 1 - R_{LS}$.

**Global Semantic Protection Degree** ($P_{GS}$). Following (Yue et al., 2021; Chen et al., 2023a), we adopt the *input inference attack* (Yue et al., 2021) to evaluate the global semantic privacy protection level of the perturbed LLM inputs. This attack aims to assess how well the original content of a private text can be inferred using a pre-trained model. Specifically, the attack process involves

replacing each token in the private text with a [mask] token, one at a time. A pre-trained BERT model (Devlin et al., 2018) is then employed to predict the original token that was masked. The success rate of this attack, denoted as $R_{GS}$, is a measure of how frequently BERT accurately predicts the masked tokens. A higher success rate indicates that the model can more easily infer the original content, suggesting lower privacy protection. Therefore, the corresponding privacy-preserving level is quantified as $P_{GS} = 1 - R_{GS}$. Since this metric relies on contextual information to infer text privacy and LLM responses in classification tasks are typically concise (e.g., 'It is positive'), it is not suitable for evaluation using this metric. Hence, we use $P_{GS}$ only to assess the privacy of the perturbed LLM inputs.

**Rouge: Recall-Oriented Understudy for Gisting Evaluation** ($U_{Rouge}$). Following (Xiao et al., 2023), we adopt $U_{Rouge_1}$, $U_{Rouge_2}$ and $U_{Rouge_L}$ (Lin, 2004) to evaluate the quality of LLM responses in the Medical Q/A dataset. The set of tokens from the LLM-generated text is represented by $Y$, and that from the ground truth text is represented by $Y_{gt}$. The number of overlapping unigrams between $Y$ and $Y_{gt}$ is denoted as $\mathcal{F}_{o1}(Y, Y_{gt})$, and for overlapping bigrams as $\mathcal{F}_{o2}(Y, Y_{gt})$. Additionally, the total number of unigrams in $Y_{gt}$ is denoted as $U(Y_{gt})$ and the total number of bigrams as $B(Y_{gt})$. The longest common subsequence shared between $Y$ and $Y_{gt}$ is represented by $\mathcal{F}_L(Y, Y_{gt})$. The formulas for $U_{Rouge_1}$, $U_{Rouge_2}$, and $U_{Rouge_L}$ are thus formulated as:

$$U_{Rouge_1} = \frac{\mathcal{F}_{o1}(Y, Y_{gt})}{U(Y_{gt})}. \tag{10}$$

$$U_{Rouge_2} = \frac{\mathcal{F}_{o2}(Y, Y_{gt})}{B(Y_{gt})}. \tag{11}$$

$$U_{Rouge_L} = \frac{\mathcal{F}_L(Y, Y_{gt})}{max(|Y|, |Y_{gt}|)}. \tag{12}$$

# 3 ADDITIONAL EXPERIMENTS

## 3.1 USER STUDY

We conduct a user study on our PrivateChat, highlighting its unique advantages over SanText (Yue et al., 2021), CusText (Chen et al., 2023a) and HaS (Chen et al., 2023b). Ten independent participants are recruited for the study. They are given 100 communication examples between users and GPT-4 (OpenAI, 2023), processed by the

Table 1: User study on our PrivateChat.

|  | SanText | CusText | HaS | PrivateChat |
|---|---|---|---|---|
| Privacy | 2.831 | 1.556 | 1.714 | **4.641** |
| Utility | 2.204 | 2.571 | 2.371 | **4.619** |

above four methods (refer to Fig.5 for the template of the communication examples). They score the methods on privacy and utility performance (scale of 1 to 5, higher is better). Results are reported in Tab. 1. Our PrivateChat outperforms the others in both two metrics, showcasing its capability in safeguarding user privacy and maintaining effective user-LLM communication.

## 3.2 ABLATION STUDY ON THE CODEBOOK OF THE SYSTEM PROMPT PERTURBATION MODEL

In our system prompt perturbation model, each code in the codebook consists of a random combination of $N_c$ ASCII characters. To assess the impact of $N_c$, we evaluate the performance changes of our PrivateChat under varying $N_c$ settings on the SST-2 dataset (Wang et al., 2018). As shown in Tab. 2, increasing $N_c$ significantly improves the privacy of the prompt but reduces the LLM's comprehension, leading to decreased accuracy.

Table 2: Ablation study on code length $N_c$ in codebook.

| Models | $P_{GS}$ | $P_{I\_LS}$ | $P_{O\_LS}$ | $U_{ACC}$ |
|---|---|---|---|---|
| $N_c = 1$ | 0.825 | 0.857 | **0.999** | **0.864** |
| $N_c = 2$ | 0.836 | 0.914 | 0.958 | 0.798 |
| $N_c = 3$ | **0.947** | **1.000** | 0.831 | 0.685 |

Additionally, we also assess the impact of the number of codes (termed as $R$) contained in the codebook on the SST-2 dataset (Wang et al., 2018). As shown in Tab. 3, if the number of codes in the codebook is too small (i.e., $R = 25$), it is difficult for the network to find the optimal replacement for the perturbation characters, thereby reducing performance. Conversely, if there are too many codes (i.e., $R = 200$), it results in an excessive

Table 3: Ablation study on codebook length $R$.

| Models | $P_{GS}$ | $P_{I\_LS}$ | $P_{O\_LS}$ | $U_{ACC}$ |
|---|---|---|---|---|
| $R = 25$ | 0.740 | 0.628 | 0.762 | 0.554 |
| $R = 50$ | 0.825 | 0.857 | **0.999** | **0.864** |
| $R = 100$ | 0.837 | **0.875** | 0.990 | 0.798 |
| $R = 200$ | **0.863** | 0.714 | 0.782 | 0.663 |

number of parameters that need training, making optimization
difficult and reducing the effectiveness of the prompt. Therefore, we chose the number of codes
$R = 50$ to achieve a relatively optimal performance.

### 3.3 Ablation Study on the Perturbation Range

As the most crucial part of the private system prompt is the en-
cryption details (i.e., the encryption method and the key), we
design a variant of our PrivateChat that only perturbs the en-
cryption details instead of perturbing the entire plaintext system
prompt. As shown in Tab. 4, we test the performance changes
of this variant under varying $N_c$ settings on the SST-2 dataset
(Wang et al., 2018). It is evident that increasing $N_c$ improves

Table 4: Results of perturbing
only the encryption details.

| Models | $P_{GS}$ | $P_{I\_LS}$ | $P_{O\_LS}$ | $U_{ACC}$ |
|---|---|---|---|---|
| $N_c = 1$ | 0.722 | 0.800 | **1.000** | **0.901** |
| $N_c = 2$ | 0.800 | 0.889 | 0.998 | 0.822 |
| $N_c = 3$ | **0.864** | **1.000** | 0.987 | 0.751 |

the effectiveness of privacy protection but reduces the accuracy of LLM responses. Compared to
Tab. 2, perturbing only the encryption details, rather than the entire prompt content, leads to better
accuracy in LLM responses. However, it also makes it easier to infer privacy from the context.

### 3.4 Ablation Study on the Privacy Reward

Our privacy reward function $\mathcal{R}_p$ consists of a semantic-level
difference function $\mathcal{F}_{sem}$ and a character-level difference func-
tion $\mathcal{F}_{char}$, where each difference function calculates on both
global content (the entire private system prompt) and local con-
tent (the encryption details portions of the prompt). To demon-
strate the effectiveness of this design, we compare our method
with four variants: (**i**) Privacy reward without semantic-level

Table 5: Ablation study on the pri-
vacy reward.

| Models | $P_{GS}$ | $P_{I\_LS}$ | $P_{O\_LS}$ | $U_{ACC}$ |
|---|---|---|---|---|
| PrivateChat(w/o $\mathcal{F}_{sem}$) | 0.771 | 0.833 | 0.970 | 0.857 |
| PrivateChat(w/o $\mathcal{F}_{char}$) | 0.816 | 0.572 | 0.949 | 0.822 |
| PrivateChat(w/o $\mathcal{F}_{loc}$) | 0.708 | 0.571 | **0.999** | **0.867** |
| PrivateChat(w/o $\mathcal{F}_{glob}$) | 0.774 | 0.853 | 0.997 | 0.854 |
| PrivateChat | **0.825** | **0.857** | **0.999** | 0.864 |

function (w/o $\mathcal{F}_{sem}$), (**ii**) Privacy reward without character-level function (w/o $\mathcal{F}_{char}$), (**iii**) Privacy
reward without local reward (w/o $\mathcal{F}_{loc}$) and (**iv**) Privacy reward without global reward (w/o $\mathcal{F}_{glob}$)
on the SST-2 dataset (Wang et al., 2018). As shown in the Tab. 5, our method achieves the best
privacy and utility relative to these four variants, showing the effectiveness of each component.

### 3.5 Ablation Study on the Black-box Optimization Framework

We assess the impact of the number $J$ of
sampled perturbation vectors on the SST-2
dataset (Wang et al., 2018). Fig.2 (a) dis-
plays the privacy and utility performance of
our PrivateChat under different $J$ values.
Fig.2 (b) shows the number of API calls
required for the model to converge under
different sampling numbers $J$. The results
show that as $J$ increases, utility and privacy

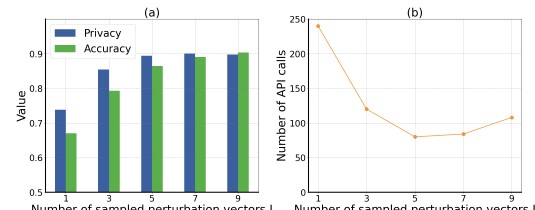

Figure 2: Ablation Study on the sampling number $J$.

improve. When $J$ exceeds 5, the improvements become marginal in Fig.2 (a) while the number of
API calls required for optimization increased in Fig.2 (b). Therefore, to achieve better results with a
small number of training samples, we choose $J = 5$ in our experiments.

### 3.6 Case Study

In Fig. 3, Fig. 4 and Fig. 5, we give more communication examples with GPT-4 (OpenAI, 2023) on
SST-2 (Wang et al., 2018), QNLI (Wang et al., 2018) and medical Q/A (Liu et al., 2023) datasets,
respectively. It is obvious that differential privacy methods (i.e., CusText (Chen et al., 2023a) and
SanText (Yue et al., 2021)) induce incoherence in text, thereby complicating the processing and
accurate response generation by LLMs. Anonymization method (i.e., HaS (Chen et al., 2023b)) only
protects partial privacy entities and destroys the original meaning of the user queries and become
ineffective when the task is related to the protected privacy entities, as shown in Fig. 4 and Fig. 5.
In contrast, our PrivateChat does not cause information loss or disrupt the original semantics of
user queries, and it enables privacy protection at both the user input and LLM output ends (i.e.,
end-to-end encrypted communication).

| Method | | SST-2 dataset |
|---|---|---|
| **PlainText** | System prompt: | Answer the question correctly: |
| | User query: | Based on the emotional tendencies of the following sentences, determine whether it is positive or negative. Question: moore 's performance impresses almost as much as her work with haynes in 1995 's safe. Answer: It is (positive/negative) |
| | LLM response: | It is positive. |
| **CusText+** | System prompt: | Answer the question correctly: |
| | User query: | Based on the emotional tendencies of the following sentences, determine whether it is positive or negative. Question: mor 's shows infuriates throughout iike considerably iike sa collaborate utilizes scholz pour 1999 's assured. Answer: It is (positive/negative) |
| | LLM response: | The provided text does not contain clear emotional tendencies or explicit sentiments that would allow me to determine whether it is positive or negative. |
| **SanText** | System prompt: | Answer the question correctly: |
| | User query: | Based on the emotional tendencies of the following sentences, determine whether it is positive or negative. Question: mimics gluing performance oeuvre planned singularly much as her 270 with haynes in 1790 impression safe. Answer: It is (positive/negative) |
| | LLM response: | It is negative. |
| **HaS** | System prompt: | Answer the question correctly: |
| | User query: | Based on the emotional tendencies of the following sentences, determine whether it is positive or negative. Question: Johnson's performance impresses almost as much as her work with Smith in 1995's safe. Answer: It is (positive/negative) |
| | LLM response: | It is positive. |
| **Our PrivateChat** | System prompt: | We w0ll us6 ad_@crhd3=en@rjZptioXl s)Y3Lnderd v^Q; J~ommu{L-_cate. I uill prsfiide the k{y, iriti9{:v:ctor and eM{ryp9id ouestiKn )ncoded in besSTs{xeyof{Tr. |
| | | (Plaintext: We will use advanced encryption standard to communicate. I will provide the key, initial vector and encrypted question encoded in base sixty-four.) |
| | User query: | 7UYaNYaddPXmmZhvbuCIuQ==JhebvgRRedpw7Ej6kH0dKA==pD1q/aOyAcjla4b seI9zYF8jD0crdiE6btoncst5ORRRVVaIi5DPUIJJ4/iuslKSIVkHRZHGYV+1jBg4ssI EgTbKX7zigKB+Rzcoe4eGsHNk8qUpvQgtjBUfdSZ5Dy0xjtKIPnIlwc91pLLL2fYT RnnJwZRHumsEYIIMCjZ1HZf4rSnDI0wNAtSKjDtkYq+nwLcqfFF5QUQGcRh+X z2H0LvnNy9QzLZktcVxixUKb7iVoCAIuJFIZ4+wGVmXbvdmw2ARK57eOxsRkK ysIGn3SkKYgJiONASkri6i2Kp37DRMXvgMKWpZcDyFElHPY+3d |
| | LLM response: | I5f0DzipDS20/vsSO524vg== |
| | Decrypted response: | It is positive. |

Figure 3: Communication examples with GPT-4 (OpenAI, 2023) on the SST-2 dataset (Wang et al., 2018).

## 3.7 DISCUSSION

Here, we provide more discussion of our ablation studies in the main text and the societal impact of our work. As shown in Tab.2 within the main content, our learnable private system prompt outperforms those prompts generated by the differential privacy method (DP-based Prompt) and the anonymization method (Anon-based Prompt). The possible reasons are: (i) The DP method and anonymization method can only perturb or replace partial words in the plaintext prompt, which increases the risk of attackers inferring encrypted details from the remaining unchanged words, thereby reducing privacy. (ii) Since they complete the privatization process before sending the system prompts to the LLM and do not use LLM feedback to adjust parameters, they cannot ensure that the LLM can effectively understand the generated private prompts. Moreover, their word-level perturbation or replacement further disrupts the coherence of the overall sentence, leading to the LLM's inability to process and thus failing to generate accurate responses, resulting in reduced utility. Tab.2 within the main content also shows that our character-level perturbation strategy has advantages over word-level and token-level perturbation strategies. The potential reasons include: (i) Character-level perturbation strategy allows for finer-grained modifications to the text without disrupting the basic structure and grammar of the prompt. Therefore, the model can still understand and process these modified prompts, generating accurate responses. (ii) Word-level and token-level perturbation strategies may miss some sensitive words, allowing attackers to infer key information from the remaining parts. Character-level perturbations can more comprehensively obscure sensitive content, reducing the risk of information leakage.

As shown in Tab.3 within the main content, our SE-SPSA optimizer achieves the best performance compared with other black-box optimizers. Random search (Bergstra & Bengio, 2012) performs poorly because it does not consider the results of previous evaluations, resulting in a very low probability of sampling near-optimal solutions, especially in high-dimensional parameter spaces. Similarly, the exploration strategy used by the reinforcement learning method (e.g., DDPG (Lillicrap et al., 2015)) is insufficient to effectively explore the potential solution space, resulting in high training time and costs. The one-sided gradient optimization method (e.g., BAR (Tsai et al., 2020)), which has fixed perturbation directions, is susceptible to noise in high-dimensional spaces, leading to instability and inaccuracy. Although SPSA (OpenAI, 2023) is effective for optimizing high-dimensional parameters, the process is unstable and each iteration requires multiple evaluations, making it costly for tasks that depend on expensive API calls for evaluation. Despite improvements by BlackVIP (Oh et al., 2023) in updating parameters to mitigate the impact of poor gradient estimates, it does not fundamentally address the issue of unstable convergence due to high variance in gradient estimates. Our SE-SPSA constrains this variance, aligning gradient estimates more closely with the correct gradient direction, thus stabilizing and speeding up the optimization process. This results in enhanced performance, as well as reduced training time and costs.

Tab.4 within the main content shows the performance of our method under various cloud LLMs. To demonstrate the model's generalization capability, we tested the results on different LLMs with the private system prompt trained on GPT-4 (OpenAI, 2023). As a result, the privacy of the input (i.e., $P_{GS}$ and $P_{I\_LS}$) remains unchanged, while the privacy and utility of the output (i.e., $P_{O\_LS}$ and $U_{ACC}$) vary. However, our method consistently demonstrated strong privacy and robust utility and significantly outperforms other methods.

Our method has certain positive societal impacts. The privacy issues associated with cloud LLMs have always been a concern in society, and some countries and governments have even banned their use due to privacy leaks from these large models. Our approach introduces a new paradigm for secure communication with cloud LLMs, helping users protect their privacy in an effective and economical way without compromising normal usage. Our method is also an inspirational work, bringing potential inspiration and contributions to future privacy protection efforts.

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

| Method | | QNLI dataset |
|---|---|---|
| **PlainText** | System prompt: | Answer the question correctly: |
| | User query: | Task: determine if the context sentence contains the answer to the question. respond with "entailment" if the sentence contains the answer, or "not entailment" if it does not. Question: 'Which network broadcasted Super Bowl 50 in the U.S.?' Answer: 'CBS broadcast Super Bowl 50 in the U.S., and charged an average of $5 million for a 30-second commercial during the game.' Does the statement contains the answer to the question? output: (entailment/not entailment) |
| | LLM response: | entailment |
| **CusText+** | System prompt: | Answer the question correctly: |
| | User query: | Task: determine if the context sentence contains the answer to the question. respond with "entailment" if the sentence contains the answer, or "not entailment" if it does not. Question: 'Which web emits Super Bowl 874 in the U.S.?' Answer: 'CBS broadcasted Super Bowl 439 in the U.S., and impeached an averages of $5 mio for a 30-second merchant during the game.' Does the statement contains the answer to the question? output: (entailment/not entailment) |
| | LLM response: | not entailment |
| **SanText** | System prompt: | Answer the question correctly: |
| | User query: | Task: determine if the context sentence contains the answer to the question. respond with "entailment" if the sentence contains the answer, or "not entailment" if it does not. Question: 'Which network Danko Hayes Bowl 50 in Product U.S?' Answer: 'CBS broadcast Super Oldfield 50 Nene the U.S. , celibacy charged thru average of $ 5 million for a 30 - second partnering during the Worldwide.' Does the statement contains the answer to the question? output: (entailment/not entailment) |
| | LLM response: | not entailment |
| **HaS** | System prompt: | Answer the question correctly: |
| | User query: | Task: determine if the context sentence contains the answer to the question. respond with "entailment" if the sentence contains the answer, or "not entailment" if it does not. Question: 'Which network broadcasted World Cup Final in the UK?' Answer: 'ABC broadcast UEFA Champions League in the UK., and charged an average of $7 million for a 45-second commercial during the match.' Does the statement contains the answer to the question? output: (entailment/not entailment) |
| | LLM response: | not entailment |
| **Our PrivateChat** | System prompt: | Un&$r@rstaP#dLFmC^ enO_ury;fted quhFstio*OO; Enc2rypLj yo#Pr a@_swer usin<2 a?$c#2esO.n| c@X[+?je/b with af;8seFc c+f tc8rV$e |
| | | (Plaintext: Understand my encrypted question. Encrypt your answer using a Caesar cipher with a key of three) |
| | User query: | Wdvn: ghwhuplqh li wkh frqwhaw vhqwhqfh frqwdlqv wkh dqvzhu wr wkh txhvwlrq. uhvsrqg zlwk "hqwdlophqw" li wkh vhqwhqfh frqwdlqv wkh dqvzhu, ru "qrw hqwdlophqw" li lw grhv qrw. Txhvwlrq: 'Zklfk qhwzrun eurdgfdvwhg Vxshu Erzo 50 lq w-kh X.V.?' Dqvzhu: 'FEV eurdgfdvw Vxshu Erzo 50 lq wkh X.V., dqg fkdujhg dq dyh-udjhri $5 ploolrq iru d 30-vhfrqg frpphufldo gxulqj wkh jdph.' Grhv wkh vwdwhphq-wfrqwdlqv wkh dqvzhu wr wkh txhvwlrq? rxwsxw: (hqwdlophqw/qrw hqwdlophqw) |
| | LLM response: | irxempqirx |
| | Decrypted response: | entailment |

Figure 4: Communication examples with GPT-4 (OpenAI, 2023) on the QNLI dataset (Wang et al., 2018).

| Method | | Medical Q/A dataset |
|---|---|---|
| **PlainText** | System prompt: | Answer the question correctly: |
| | User query: | What causes Ewing's sarcoma? |
| | LLM response: | Ewing's sarcoma is primarily caused by a genetic mutation, specifically a translocation between chromosomes 11 and 22, which creates an abnormal fusion gene known as EWSR1-FLI1. |
| **CusText+** | System prompt: | Answer the question correctly: |
| | User query: | What reason Ewing's sarcoma? |
| | LLM response: | Ewing's sarcoma is primarily caused by genetic mutations, specifically translocations between chromosomes 11 and 22. |
| **SanText** | System prompt: | Answer the question correctly: |
| | User query: | teen causes Brugada' sarcoma? |
| | LLM response: | It seems there might be a misunderstanding or confusion in your question. |
| **HaS** | System prompt: | Answer the question correctly: |
| | User query: | What causes Johnson's sarcoma? |
| | LLM response: | The specific causes of each individual's condition are not provided in the question. |
| **Our PrivateChat** | System prompt: | WA wiDl Nse adveS8ed en@ryttiBn stan>aGdg(> c#mqnicate. I 3illp rovide Jhe ke[, injtial ve0toI and encr(rIed qulQtion ndcodqd un kaee teiCry-de3. |
| | | (Corresponding plaintext: We will use advanced encryption standard to communicate. I will provide the key, initial vector and encrypted question encoded in base thirty-two.) |
| | User query: | 5VDBUNMGTV2PLZUZTBXW5YEIXE======EYLZXPQEKF45U4HMJD5JA7I5FA======55T2QS6QZB3LI766BTA64SDVXUDJNCRM35QEHDMX5DLHSLPHK72Q==== |
| | LLM response: | JLOGAK5BTXHK37Y4CLF4VSCZKUUKWX7APQ6YRZD3D6ASL25424EQ4FSWIZDW2PDTKA73YFQO52ELZVPJIWRKPA3KY27HKMLHWZF6SJLRYSOEOKX3GGCMZOTCN3KHMMPLQXSP6O4RMMXFOCDNZKAEUPQEBMOISRYQEAJOSNOXDIIDFIJWMEOA==== |
| | Decrypted response: | Ewing's sarcoma is caused by a genetic anomaly involving a translocation between chromosomes 11 and 22. |

Figure 5: Communication examples with GPT-4 (OpenAI, 2023) on the medical Q/A dataset (Liu et al., 2023).

Yu Chen, Tingxin Li, Huiming Liu, and Yang Yu. Hide and seek (has): A lightweight framework for prompt privacy protection. *arXiv preprint arXiv:2309.03057*, 2023b.

Jacob Devlin, Ming-Wei Chang, Kenton Lee, and Kristina Toutanova. Bert: Pre-training of deep bidirectional transformers for language understanding. *arXiv preprint arXiv:1810.04805*, 2018.

Timothy P Lillicrap, Jonathan J Hunt, Alexander Pritzel, Nicolas Heess, Tom Erez, Yuval Tassa, David Silver, and Daan Wierstra. Continuous control with deep reinforcement learning. *arXiv preprint arXiv:1509.02971*, 2015.

Chin-Yew Lin. Rouge: A package for automatic evaluation of summaries. In *Text summarization branches out*, pp. 74–81, 2004.

Fenglin Liu, Tingting Zhu, Xian Wu, Bang Yang, Chenyu You, Chenyang Wang, Lei Lu, Zhangdaihong Liu, Yefeng Zheng, Xu Sun, et al. A medical multimodal large language model for future pandemics. *NPJ Digital Medicine*, 6(1):226, 2023.

Changdae Oh, Hyeji Hwang, Hee-young Lee, YongTaek Lim, Geunyoung Jung, Jiyoung Jung, Hosik Choi, and Kyungwoo Song. Blackvip: Black-box visual prompting for robust transfer learning. In *Proceedings of the IEEE/CVF Conference on Computer Vision and Pattern Recognition*, pp. 24224–24235, 2023.

OpenAI. Gpt-4 technical report. `https://cdn.openai.com/papers/gpt-4.pdf`, 2023.

Chen Qu, Weize Kong, Liu Yang, Mingyang Zhang, Michael Bendersky, and Marc Najork. Natural language understanding with privacy-preserving bert. In *Proceedings of the 30th ACM International Conference on Information & Knowledge Management*, pp. 1488–1497, 2021.

Meng Tong, Kejiang Chen, Yuang Qi, Jie Zhang, Weiming Zhang, and Nenghai Yu. Privinfer: Privacy-preserving inference for black-box large language model. *arXiv preprint arXiv:2310.12214*, 2023.

Yun-Yun Tsai, Pin-Yu Chen, and Tsung-Yi Ho. Transfer learning without knowing: Reprogramming black-box machine learning models with scarce data and limited resources. In *International Conference on Machine Learning*, pp. 9614–9624. PMLR, 2020.

Alex Wang, Amanpreet Singh, Julian Michael, Felix Hill, Omer Levy, and Samuel R Bowman. Glue: A multi-task benchmark and analysis platform for natural language understanding. *arXiv preprint arXiv:1804.07461*, 2018.

Yijia Xiao, Yiqiao Jin, Yushi Bai, Yue Wu, Xianjun Yang, Xiao Luo, Wenchao Yu, Xujiang Zhao, Yanchi Liu, Haifeng Chen, et al. Large language models can be good privacy protection learners. *arXiv preprint arXiv:2310.02469*, 2023.

Xiang Yue, Minxin Du, Tianhao Wang, Yaliang Li, Huan Sun, and Sherman SM Chow. Differential privacy for text analytics via natural text sanitization. *arXiv preprint arXiv:2106.01221*, 2021.