# OpenReview forum: "PrivateChat: A Secure Encrypted Communication Framework with Black-box LLMs"
_ICLR.cc/2025/Conference — Submitted to ICLR 2025_

### Official Review · Reviewer_P78Y · 2024-10-20

**Soundness:** 2
**Presentation:** 3
**Contribution:** 2
**Rating:** 3
**Confidence:** 4

**Summary:**

Current LLM applications are deployed on cloud-based platform, which are sensitive to several privacy attacks during prompting and answering. This work focuses on secure communication aspect to allow user to securely query their questions to cloud LLMs while privacy of both prompts and answers are preserved.

**Strengths:**

1. Considering from secure communication perspective is a great start to protect privacy interacting with black-box LLMs. The research direction is interesting.
2. Unlike previous on black-box LLM security, PrivateChat needs much less samples due to expensive API calls.
3. This work allows users and LLMs to communicate with encrypted texts.

**Weaknesses:**

1. This work is lack of a concrete security/adversary model to discuss the ability of attacks this work should be against with.
2. This work should not consider classic encryption algorithms (e.g., Caesar) as an effective approach to provide to LLMs, since modern cryptography has proved that almost all classic methods are not cryptographically secure with modern electronic computer.
3. The prompt with encryption key is still unsafe to me with described perturbation. If the system with LLMs is hijacked, the whole mechanism is still vulnerable to adversaries.

**Questions:**

No

---

### Official Review · Reviewer_m1N5 · 2024-10-23

**Soundness:** 1
**Presentation:** 2
**Contribution:** 1
**Rating:** 1
**Confidence:** 4

**Summary:**

The authors introduce a method for interacting with cloud-based LLM APIs by embedding encryption instructions into the LLM prompt. These instructions enable decryption of user queries and encryption of LLM responses using a specified encryption method and key. To keep the encryption details hidden, the authors propose a private system prompt— a perturbed version of the original system prompt that preserves the LLM’s behavior. This perturbation must preserve the meaning of the prompt for the LLM, but at the same time conceal the instructions to the service provider. This perturbation is learned through a novel black-box optimization technique called SE-SPSA, an adaptation of Simultaneous Perturbation Stochastic Approximation. The authors evaluate their and related approach on several cloud LLMs and tasks.

**Strengths:**

- Covers an important problem: secure inference of LLMs
- Proposes a baseline-based variance reduction technique for SPSA which seems to be applicable more generally

**Weaknesses:**

- The paper lacks a well-defined threat model, making it unclear whether this system provides any meaningful privacy guarantees. It appears that the system's security heavily relies on the assumption that the attacker is unaware of the encryption method. For instance, if the attacker knows that the user query in Figure 4 includes an AES key, an initialization vector, and the encrypted query, they could reconstruct the original query. This approach contradicts Kerckhoffs's principle and instead relies on security through obscurity. Even if we assume the service provider cannot determine the encryption method, other vulnerabilities exist. For instance, the provider could retain the LLM's state after a user interaction and, through further interactions, potentially reveal the decrypted user query. Given this issue, it is unclear to me whether the system provides any meaningful solution to the problem of secure inference of LLMs.
- The paper’s notion of privacy is weak. It assesses privacy by measuring the similarity between the "encrypted" input and output in an embedding space, but this definition lacks meaningful protection. Since the encrypted input also includes the encryption key, it becomes trivial to perfectly recover the plaintext, even though the encrypted input scores highly on the proposed privacy metric. The paper would benefit from adopting a more rigorous, cryptographic notion of privacy for encryption schemes such as semantic security.

**Questions:**

- What are the capabilities of the service provider in your threat model?
- Is the key part of the system prompt or in the user query? In the example in Figure 4 in the paper and Figure 5 in the supplementary material, the AES key is in the user query, but for the Caesar example and in the text, the key is in the private prompt.
- How is the LLM decrypting the input and encrypting the output? Is this done through some external code execution module that the LLM can interact with?

---

### Official Review · Reviewer_GGeX · 2024-11-02

**Soundness:** 3
**Presentation:** 2
**Contribution:** 2
**Rating:** 6
**Confidence:** 2

**Summary:**

This paper introduces a framework named “PrivateChat” aimed at addressing privacy protection issues between users and cloud-hosted black-box large language models (LLMs). The novelty of the paper lies in enabling interaction with cloud LLMs through user-customized encryption methods, while also developing an efficient black-box optimizer—Sample-Efficient Simultaneous Perturbation Stochastic Approximation (SE-SPSA)—for optimizing system prompts.

**Strengths:**

The paper proposes a new private communication framework for interacting with cloud-hosted black-box LLMs, filling a gap where existing privacy-preserving techniques struggle to adapt to black-box models. The introduction of the SE-SPSA optimizer is an interesting and effective approach to solve black-box model optimization, achieving good gradient estimates with fewer training samples.
﻿
Experimental Validation: The paper conducts extensive experiments on multiple benchmark datasets (e.g., SST-2, QNLI, Medical Q/A), validating the advantages of PrivateChat in terms of balancing privacy protection and utility preservation. Experimental results show that the proposed method performs well across different encryption algorithms (e.g., Caesar, DES, AES, ChaCha20) and is applicable to various cloud LLMs (e.g., GPT-4, GPT-4V, Sonnet, Opus), indicating its generalizability.
﻿
Integration of Theory and Experiments: The proposed system prompt perturbation module employs a character-level perturbation strategy to effectively guide LLMs without revealing encryption details. Compared to other methods, this strategy better balances privacy and utility. The baseline selection strategy in SE-SPSA employs an innovative variance reduction method, which significantly improves the stability and convergence speed of the optimization process, thereby reducing costly API calls.

**Weaknesses:**

While the paper proposes a system prompt perturbation module to conceal encryption details, its security evaluation is relatively weak, lacking in-depth discussion and experimental analysis of potential attack scenarios.

**Questions:**

Security Evaluation: Strengthen the security analysis of the system prompt perturbation module, particularly through experimental evaluations of potential inference attacks and other types of attacks, to comprehensively verify its privacy-preserving capabilities.

---

### Official Review · Reviewer_UrJc · 2024-11-02

**Soundness:** 3
**Presentation:** 3
**Contribution:** 2
**Rating:** 6
**Confidence:** 4

**Summary:**

The paper introduces PrivateChat, a novel private communication framework designed to allow users to interact securely with cloud-based large language models (LLMs) through user-specific encryption techniques. The framework’s core innovation is a private system prompt generator that instructs the cloud LLM to interpret and respond to user queries according to the specified encryption methods. The authors trained this prompt generator using an efficient black-box optimization algorithm. This work is a pioneering approach to enabling encrypted interactions with cloud-based LLMs, and it provides a comprehensive discussion on both model utility and privacy protection aspects.

**Strengths:**

The paper proposes a relatively novel solution to address privacy concerns in communication between LLMs and users, presenting this solution in a relatively clear and complete manner. Throughout the paper, the authors address most of the concerns I had, providing thoughtful explanations. Additionally, this work is likely to be of value for future research in related areas. Overall, I find the paper’s originality and clarity to be relatively strong.

**Weaknesses:**

The proposed solution in the paper is relatively simple and direct; however, I have some concerns regarding the communication efficiency of the framework presented. For instance, the authors do not conduct experimental discussions on communication overhead and efficiency, nor do they provide comparisons with other methods. Additionally, the paper lacks a discussion on the key lengths associated with different encryption methods, which could help clarify how the framework balances privacy protection and usability. Addressing these two points may enhance the integrity of the work.

**Questions:**

Could the authors include a discussion on communication efficiency? Additionally, it would be helpful to examine how the model’s efficiency varies under the same encryption method but with different key lengths. I believe that addressing these aspects would contribute positively to the completeness of the paper.

---

### Meta-Review · Area_Chair_ptau · 2024-12-20

**Metareview:**

This paper introduces PrivateChat, a novel framework that enables secure communication between users and cloud-hosted large language models (LLMs) using user-specific encryption methods, such as AES. Addressing concerns about privacy leakage in LLM applications, PrivateChat ensures that user queries and responses are encrypted during interactions while concealing encryption details from potential attackers.

At the core of PrivateChat is a private system prompt that instructs the cloud LLM to process and respond in encrypted text according to the user's chosen encryption method. To optimize this system prompt efficiently, the authors propose a Sample-Efficient Simultaneous Perturbation Stochastic Approximation (SE-SPSA) algorithm. SE-SPSA is a black-box optimization technique designed to minimize API calls while ensuring effective and economical training. It integrates a baseline-based variance reduction strategy with Simultaneous Perturbation Stochastic Approximation to enhance efficiency.

Experiments on benchmark datasets demonstrate that PrivateChat achieves secure and reliable communication across various encryption methods. It effectively balances privacy protection with model utility, preserving the intended functionality of cloud-based LLMs while safeguarding sensitive user information. This pioneering approach addresses critical privacy concerns and sets a foundation for encrypted interactions with LLM APIs.

The reviewers have provided two major criticisms. First, the attack/threat model is not well-defined. Secondly, the empirical evaluation and analysis are insufficient and lack discussions and analysis.

**Additional Comments On Reviewer Discussion:**

The authors did not participate in the rebuttal process.

---

### Decision · Program_Chairs · 2025-01-22

Reject